# LONG TERM FAIRNESS VIA PERFORMATIVE DISTRIBUTIONALLY ROBUST OPTIMIZATION

## ABSTRACT

Fairness researchers in machine learning (ML) have coalesced around several fairness criteria which provide formal definitions of what it means for an ML model to be fair. However, these criteria have some serious limitations. We identify four key shortcomings of these formal fairness criteria and address them by extending performative prediction to include a distributionally robust objective. Performative prediction is a recent framework developed to understand the effects of when deploying a model influences the distribution on which it is making predictions. We prove a convergence result for our proposed repeated distributionally robust optimization (RDRO). We further verify our results empirically and develop experiments to demonstrate the impact of using RDRO on learning fair ML models.

## 1 INTRODUCTION

In the past two decades, machine learning (ML) has moved from the confines of research institutes and university laboratories to become a core element of the global economy. ML models are now deployed at enormous scales in complex environments, often making high stakes decision. Too often, however, this is done without adequate concern for the fairness and robustness of these ML models. Fairness in ML is a burgeoning research area, but much of the work in fairness, particularly in defining formal fairness criteria, has been limited to the static classification setting.

Efforts to define fairness in ML have resulted in myriad criteria being proposed, many of which are equivalent to, or relaxations of, three core definitions of fairness: *independence, separation,* and *sufficiency* Barocas et al. (2019); Chouldechova (2017); Corbett-Davies et al. (2017); Dwork et al. (2012); Hardt et al. (2016b); Berk et al. (2021); Zafar et al. (2017); Kleinberg et al. (2017); Woodworth et al. (2017). These formal fairness criteria assume a *sensitive characteristic* or *protected demographic group* for whom we want to ensure our model is non-discriminatory. The fairness criteria are then properties of the joint distribution of this characteristic, the output of the classifier, and the true labels of the data.

While these fairness definitions have been a useful starting point in the consideration of discrimination by ML models, they have several limitations. 1. They are not equivalent and, in most scenarios, they are incompatible. 2. They apply only to static supervised learning problems and ignore the dynamic environments characteristic of many real world scenarios with fairness concerns. 3. They rely on having access to demographic information. The definitions can only be used if one has access to the sensitive characteristic, which is often not the case. 4. They ignore intersectionality. The criteria do not take into account individuals who may sit at the intersection of several sensitive demographic groups. Fairness is fundamentally a philosophical and political question, and the notion of having a single, universal formal definition of fairness for ML is likely naïve. For this reason, this work does not attempt to formally define fairness and largely ignores the first problem noted above. We do, however, attempt to address issues 2, 3, and 4 by drawing upon two recent areas of research with implications for fairness in ML: *performative prediction* and *distributionally robust optimization* (DRO).

DRO offers a compelling and flexible method for training non-discriminatory algorithms without needing access to demographic information. Performative prediction, on the other hand, attempts to outline a theoretical framework through which we can reason about ML models in dynamic environments, when the act of deploying a model influences the distribution on which it is making decisions.

We combine these two areas of research to extend the performative prediction framework developed in Perdomo et al. (2020). The work in performative prediction has thus far only concerned *risk minimization* and *empirical risk minimization* (ERM), so we extend definitions to include a distributionally robust objective and prove an analogous convergence result to that shown in Perdomo et al. (2020). Due to space constraints, we provide an extended discussion of related work in section A.1 of the appendix.

## 2 BACKGROUND

### 2.1 DISTRIBUTIONALLY ROBUST OPTIMIZATION

The *de facto* objective used in most supervised learning settings is ERM. In ERM we attempt to approximate minimization of the expected loss over the data generating distribution by minimizing the average loss over our data set:

$$\hat{h} = \arg\min_{h \in \mathcal{H}} \frac{1}{n} \sum_{i=1}^{n} \ell(h(x_i), y_i),$$

where $h$ represents a hypothesis or model, $\mathcal{H}$ a hypothesis class, and $\ell$ a non-negative loss function. ERM is intuitively appealing and has important theoretical guarantees associated with it Vapnik (1991). It can, however, be problematic when it comes to fairness concerns. Since we are averaging the loss over our data points, in general, ERM causes an algorithm to focus on majority cases while ignoring minority cases or rare events.

DRO, on the other hand, considers the *distributionally robust* problem in which we construct an *uncertainty set* around the data generating distribution and attempt to minimize the expected loss on the worst-case distribution within this uncertainty set. Following Duchi & Namkoong (2021) we define our uncertainty set as

$$\mathcal{U}_f(P) = \{Q : D_f(Q||P) \leq \rho\},$$

where $D_f(Q||P)$ is an $f$-divergence between probability distributions $Q$ and $P$.

Formally the distributionally robust problem is as follows:

$$\underset{\theta \in \Theta}{\text{minimize}} \left\{ \sup_{Q \ll P_0} \{\mathbb{E}_Q[\ell(\theta; X)] : Q \in \mathcal{U}_f(P_0)\} \right\},$$

where $\Theta \subset \mathbb{R}^d$ is the parameter (model) space, $P_0$ is the data generating distribution on the measure space $(\mathcal{X}, \mathcal{A})$, $X$ is a random element of $\mathcal{X}$ and $\ell : \Theta \times \mathcal{X} \to \mathbb{R}$ is a loss function.

In this formulation of DRO, the uncertainty set is determined by an $f$-divergence between $Q$ and $P_0$ and $\{\mathbb{E}_Q[\ell(\theta; X)] : Q \in \mathcal{U}_f(P_0)\}$ is the set of all expected losses over the $f$-divergence ball of radius $\rho$, centred at $P_0$. Alternative DRO formulations utilizing different measures of distance between probability distributions such as Wasserstein balls have also been explored Wald (1945); Wozabal (2012); Pflug & Wozabal (2007); Lee & Raginsky (2018).

Unlike ERM, DRO does not equally weight each data point, but instead up-weights data points on which the model is achieving high loss. This means that the model should achieve somewhat uniform performance on individuals across demographic groups Duchi & Namkoong (2021); Duchi et al. (2020); Namkoong & Duchi (2016); Hashimoto et al. (2018). Note that the distributionally robust objective does not require any access to demographic information and, at least potentially, naturally accounts for intersectionality.

One can optimize the distributionally robust objective directly via the primal form, specified above, or alternatively through a dual formulation. For convex losses, the dual form is jointly convex in the parameters of the model and the dual variables Duchi & Namkoong (2021). A detailed discussion of the dual formulation can be found in A.2. The dual formulation also helps provide intuition for why DRO is more likely to result in fair ML models.

### 2.2 PERFORMATIVE PREDICTION

Performative prediction attempts to formalize the notion of a model affecting the distribution on which it is making predictions in a type of feedback loop. There are many examples of scenarios with

fairness concerns in which this is likely to occur, such as predictive policing or college admission decisions.

This notion of performativity of an algorithm is captured by a *distribution map*, $\mathcal{D}(\theta)$, which maps the data generating distribution to a new distribution as a function of the parameters of the model which has been deployed. In performative prediction, the problem of risk minimization becomes *performative risk* minimization and involves minimizing the expected loss on the induced distribution rather than the data generating distribution. The performative risk of a model is defined as

$$PR(\theta) = \mathbb{E}_{Z \sim \mathcal{D}(\theta)}[\ell(Z; \theta)].$$

Solution concepts differ for performative prediction and traditional supervised learning, as the distribution shift complicates the learning problem. To capture this distinctive problem, Perdomo et al. (2020) define *performative optimality* and *performative stability*.

**Definition 2.1.** (performative optimality) A model, $f_{\theta_{PO}}$, is performatively optimal if the following relationship holds:

$$\theta_{PO} = \arg\min_{\theta \in \Theta} \mathbb{E}_{Z \sim \mathcal{D}(\theta)}[\ell(Z; \theta)].$$

Equivalently, $\theta_{PO} = \arg\min_{\theta \in \Theta} PR(\theta)$ where $PR(\theta)$ is the performative risk as defined above.

A performatively optimal point is a minimizer of the performative risk. An alternative solution concept is referred to as *performative stability*.

**Definition 2.2.** (performative stability) A model, $f_{\theta_{PS}}$, is performatively stable if the following relationship holds:

$$\theta_{PS} = \arg\min_{\theta \in \Theta} \mathbb{E}_{Z \sim \mathcal{D}(\theta_{PS})}[\ell(Z; \theta)].$$

Define $DPR(\theta, \theta') := \mathbb{E}_{Z \sim \mathcal{D}(\theta)}[\ell(Z; \theta')]$ as the decoupled performative risk; then $\theta_{PS} = \arg\min_{\theta \in \Theta} DPR(\theta_{PS}, \theta)$.

A performatively stable model is not necessarily a minimizer of the performative risk, but it is optimal on the distribution it induces.

Performative prediction presents a special case of learning under distribution drift, where the distribution drift is a function of the model that was deployed. A common approach in supervised learning under distribution drift is to retrain a model on newly collected data. While this does not directly minimize performative risk, it is a realistic representation of the strategy taken by many machine learning practitioners and can be a reasonable solution in some scenarios.

**Definition 2.3.** (repeated risk minimization) Repeated risk minimization (RRM) refers to the procedure where, starting from an initial model $f_{\theta_0}$, we perform the following sequence of updates for every $t \geq 0$:

$$\theta_{t+1} = G(\theta_t) := \arg\min_{\theta \in \Theta} \mathbb{E}_{Z \sim \mathcal{D}(\theta_t)}[\ell(Z; \theta)].$$

In Perdomo et al. (2020) the authors show that under certain smoothness and convexity conditions on the loss function and distribution map, RRM is a contraction mapping that converges to a fixed point which is a performatively stable model. The smoothness condition on the distribution map is called $\epsilon$-sensitivity and ensures that the distribution shift is relatively small for small changes in the model parameters. Section A.3 contains an extended discussion of performative prediction and formally defines all the relevant concepts.

## 3 CONVERGENCE OF REPEATED DISTRIBUTIONALLY ROBUST OPTIMIZATION

In order to extend the performative prediction framework to encompass DRO, we need to redefine many of the concepts from Perdomo et al. (2020) to account for the distributionally robust objective. The supremum over the uncertainty set introduces additional complications necessitating novel definitions and an altered proof for convergence to a performatively stable model. We define *robust performative risk*, *robust performative optimality*, *robust performative stability*, and *repeated distributionally robust optimization* (RDRO).

We begin by redefining performative risk and performatively optimal and stable models in terms of a robust objective. In the following $\mathcal{D}(\theta)$ is a distribution map and $D_f(Q||\mathcal{D}(\theta))$ is an $f$-divergence ball of radius $\rho$.

**Definition 3.1.** (robust performative risk) The robust performative risk of a model, $\theta$, is

$$\text{RPR}(\theta) = \sup_{Q \ll \mathcal{D}(\theta)} \{\mathbb{E}_{Z \sim Q}[\ell(Z;\theta)] : D_f(Q||\mathcal{D}(\theta)) \leq \rho\}.$$

Robust performative risk differs from performative risk in that the induced distribution, $\mathcal{D}(\theta)$, is now the centre of an $f$-divergence ball over which a supremum of the expected loss is taken.

**Definition 3.2.** (robust performative optimality) A model $f_{\theta_{PO}}$ is robustly performatively optimal if the following relationship holds:

$$\theta_{PO} = \arg\min_{\theta \in \Theta} \sup_{Q \ll \mathcal{D}(\theta)} \{\mathbb{E}_{Z \sim Q}[\ell(Z;\theta)] : D_f(Q||\mathcal{D}(\theta)) \leq \rho\}.$$

Equivalently, $\theta_{PO} = \arg\min_{\theta \in \Theta} \text{RPR}(\theta)$ where $\text{RPR}(\theta)$ is the robust performative risk as defined above.

**Definition 3.3.** (robust performative stability) A model $f_{\theta_{PS}}$ is robustly performatively stable if the following relationship holds:

$$\theta_{PS} = \arg\min_{\theta \in \Theta} \sup_{Q \ll \mathcal{D}(\theta_{PS})} \{\mathbb{E}_{Z \sim Q}[\ell(Z;\theta)] : D_f(Q||\mathcal{D}(\theta_{PS})) \leq \rho\}.$$

Define $\text{RDPR}(\theta, \theta') := \sup_{Q \ll \mathcal{D}(\theta)} \{\mathbb{E}_{Z \sim Q}[\ell(Z;\theta')] : D_f(Q||\mathcal{D}(\theta)) \leq \rho\}$ as the robust decoupled performative risk; then $\theta_{PS} = \arg\min_{\theta \in \Theta} \text{RDPR}(\theta_{PS}, \theta)$.

We discussed earlier that previous work has shown that repeated risk minimization converges to a performatively stable model under certain assumptions on the loss and distribution map Perdomo et al. (2020). We can analogously define repeated distributionally robust optimization as follows.

**Definition 3.4.** (repeated distributionally robust optimization) Repeated distributionally robust optimization (RDRO) refers to the procedure where, starting from an initial model $f_{\theta_0}$, we perform the following sequence of updates for every $t \geq 0$:

$$\theta_{t+1} = G(\theta_t) := \arg\min_{\theta \in \Theta} \sup_{Q \ll \mathcal{D}(\theta_t)} \{\mathbb{E}_{Z \sim Q}[\ell(Z;\theta)] : D_f(Q||\mathcal{D}(\theta_t)) \leq \rho\}.$$

As with RRM, this is an iterative procedure where we optimize the distributionally robust objective at each time step $t$. While it is only a small departure from RRM conceptually, the DRO objective has fundamentally different mathematical properties than risk minimization making it unclear whether RDRO and RRM should exhibit similar behaviour.

The assumptions on the loss function and distribution map required for convergence of RRM do not suffice for RDRO, as the supremum over the uncertainty set alters the mathematical properties of the robust objective. We adapt these conditions to extend them to the distributionally robust objective. The original conditions from Perdomo et al. (2020) can be found in A.3.

**Definition 3.5.** (robust $\beta$-joint smoothness) We say the distributionally robust objective is robustly $\beta$-jointly smooth if for all $\theta, \theta' \in \Theta$ and $z, z' \in \mathcal{Z}$ the gradient,

$$\nabla_\theta \sup_{Q \ll \mathcal{D}(\theta)} \{\mathbb{E}_{Z \sim Q}[\ell(Z;\theta)] : Q \in \mathcal{U}_f(\mathcal{D}(\theta))\},$$

exists and is $\beta$-Lipschitz in $z$ and $\theta$.

**Definition 3.6.** (robust $\gamma$-strong convexity) We say the distributionally robust objective is robustly $\gamma$-strongly convex if for all $\theta, \theta' \in \Theta$ and $z \in \mathcal{Z}$

$$\sup_{Q \ll \mathcal{D}(\theta)} \{\mathbb{E}_{Z \sim Q}[\ell(Z;\theta)] : Q \in \mathcal{U}_f(\mathcal{D}(\theta))\}$$

is $\gamma$-strongly convex.

**Definition 3.7.** (robust $\epsilon$-sensitivity)

Let $\mathcal{D}^*(\theta) = \arg\max_{Q:Q \in \mathcal{U}_f(\mathcal{D}(\theta))} \mathbb{E}_{Z \sim Q}[\ell(Z; \theta)]$. Assume that a distribution map, $D(\cdot)$, is $\epsilon$-sensitive[1]. We say that this distribution map is robustly $\epsilon$-sensitive if there exists $\omega > 0$ such that for any $\theta, \theta' \in \Theta$

$$W_1\left(\mathcal{D}^*(\theta), \mathcal{D}^*(\theta')\right) \leq \omega W_1\left(\mathcal{D}(\theta), \mathcal{D}(\theta')\right) \leq \omega\epsilon||\theta - \theta'||,$$

where $W_1$ denotes the Wasserstein-1 distance.

We now state our convergence theorem for DRO in the performative prediction setting. This theorem says that under the conditions outlined above, repeatedly optimizing a distributionally robust objective will converge to a performatively stable model at a linear rate. The proof can be found in A.4.

**Theorem 3.8.** *Suppose that the distributionally robust objective satisfies definitions 3.5 and 3.6, and that the distribution map satisfies definition 3.7. Then the following statements are true:*

1. *$||G(\theta) - G(\theta')||_2 \leq \omega\epsilon\frac{\beta}{\gamma}||\theta - \theta'||_2$, for all $\theta, \theta' \in \Theta$*

2. *If $\omega\epsilon < \frac{\gamma}{\beta}$, the iterates $\theta_t$ of RDRO converge to a unique robustly performatively stable point $\theta_{PS}$ at a linear rate: $||\theta_t - \theta_{PS}||_2 \leq \delta$ for $t \geq \left(1 - \omega\epsilon\frac{\beta}{\gamma}\right)^{-1} \log\left(\frac{||\theta_0 - \theta_{PS}||_2}{\delta}\right)$.*

*Note that $G(\theta) := \arg\min_{\theta \in \Theta} \sup_{Q \ll \mathcal{D}(\theta)}\{\mathbb{E}_{Z \sim Q}[\ell(Z; \theta)] : Q \in \mathcal{U}_f(\mathcal{D}(\theta))\}$.*

The proof of this theorem shows that RDRO is a contraction mapping and is essentially an application of the Banach fixed point theorem. Definitions 3.6, 3.5, and 3.7 provide smoothness conditions for the mapping sufficient to ensure contraction.

The importance of this theorem is that it suggests that RDRO should exhibit similar convergence behaviour to RRM. As discussed earlier, however, training models with ERM is liable to result in models which are discriminatory towards minority subgroups, while models trained with DRO should exhibit uniform performance across different subgroups. What this implies is that RRM may converge to unfair fixed points, while RDRO should converge to fair fixed points.

# 4 EXPERIMENTS

We run experiments to empirically verify the convergence of RDRO as compared to RRM, as well as to demonstrate the utility of DRO in learning fair models in the presence of data characterized by majority and minority subgroups. Implementation details can be found in A.5.

## 4.1 RRM AND RDRO ON CREDIT EXTENSION DECISIONS

For our first experiment we reproduce an experiment from Perdomo et al. (2020) using a credit extension dataset from the Kaggle competition *Give Me Some Credit*. The dataset contains information about potential borrowers and the task is to predict whether or not an individual will default on their debt. We construct a distribution map by strategically manipulating individuals features in response to the deployed classifier Hardt et al. (2016a).

In order to add a performative element to the prediction problem, we identify 3 of the 10 features as strategic features which will be altered as a function of the parameters of our model. Strategic classification is a particular instance of performative prediction in which individuals adversarially adjust their features to maximize the likelihood of classification to the positive class. Strategic classification has been explored in relation to fairness concerns in previous work Hardt et al. (2016a); Bruckner et al. (2012); Miller et al. (2020); Milli et al. (2019); D'Amour et al. (2020); Hu et al. (2019).

As described in Perdomo et al. (2020) and Hardt et al. (2016a), we assume individuals have linear utilities $u(\theta, x) = -\langle\theta, x\rangle$ and quadratic costs $c(x', x) = \frac{1}{2\epsilon}||x' - x||_2^2$. The constant $\epsilon$ controls the cost individuals incur by altering their features. Individuals thus pay a cost to manipulate their

---

[1]The formal definition of $\epsilon$-sensitivity is given in A.3.

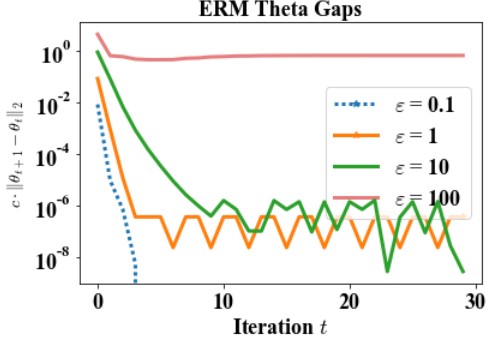
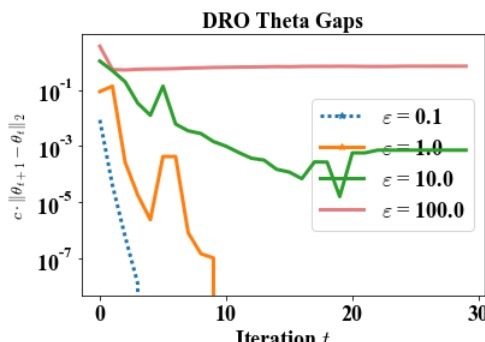

Figure 1: Plot of the normalized distance between successive values of $\theta$ for ERM.

Figure 2: Plot of the normalized distance between successive values of $\theta$ for DRO.

features in order to minimize the likelihood of the model predicting that they will default on their loan, but are unable to change the true outcome, $y \in \{0, 1\}$, of whether or not they default. Given linear utilities and quadratic costs as described here, the individuals' best response is to manipulate their features as

$$x'_S = x_S - \epsilon\theta_S,$$

where $x_S, x'_S \mathbb{R}^{|S|}$ and $|S|$ is the number of strategic features. The explanation of why this distribution map is $\epsilon$-sensitive can be found in Perdomo et al. (2020).

The procedure for updating the data according to the distribution map for strategic classification is explained in the box below.

---

**Input:** Base distribution $P$, a classifier $f_\theta$, a cost function $c$, a utility function $u$.
**Sampling procedure for** $\mathcal{D}(\theta)$**:**

1. Sample $(x, y) \sim P$
2. Compute best response $x_{BR} \leftarrow \arg\max_{x'} u(x', \theta) - c(x', x)$
3. Output sample $(x_{BR}, y)$

---

We train two logistic regression models using RRM and RDRO. The magnitude of the distribution shift is controlled by a parameter, $\epsilon$. The larger the value of $\epsilon$, the larger the distribution shift. The $\epsilon$-sensitivity condition of the distribution map is thus only satisfied for small values of $\epsilon$. Further details are given in A.6.

Figures 1 and 2 contain the distance between the learned parameters of the model on successive iterates. A distance of zero means that the model has converged to a performatively stable point. These figures show us RRM and RDRO exhibit similar convergence behaviour. Both RRM and RDRO converge for $\epsilon = 0.01$, and RDRO also converges for $\epsilon = 1.0$. Both RRM and RDRO converge to small neighbourhoods for $\epsilon = 10$, while neither converge for $\epsilon = 100$. This result empirically supports our theoretical work, suggesting that RRM and RDRO exhibit similar convergence behaviour.

## 4.2 STATIC CLASSIFICATION WITH ERM AND DRO

The credit dataset does not contain any demographic information, so to investigate the fairness properties of the fixed points resulting from RRM and RDRO we generate synthetic data, sampled from two multivariate normal distributions with different means. We refer to the majority sub-probability distribution as group A and the minority distribution as group B. Data from each individual distribution is linearly separable, but the combined distribution is not, meaning that a linear classifier must trade-off performance between the two subgroups.

To build intuition for the differing behaviour of ERM and DRO on a classification task involving subgroups and fairness concerns, we first investigate a static classification scenario, before exploring long-term fairness via performative prediction.

Our data is generated from bivariate Gaussian distributions and the label, $y$, of a given data point is 1 if the sum of its features are greater than the sum of the means of the two Gaussian distributions from which we draw samples and 0 otherwise. We have two subgroups within our data, A and B. We vary the proportion of samples from each subgroup in the experiments. The exact data generating process is given in A.6

Although this dataset is extremely simple, it is characterized by a feature that represents a central concern for fairness in machine learning, namely that the conditional probability distributions, $P(y|x)$, are significantly different for distinct subsets of the data. This example is intended to represent a simplified abstract instance of a population with majority and minority subgroups in order to see how the behaviour of ERM and DRO differ in this circumstance.

We generate three distinct datasets on which to train our algorithms, each made up of differing proportions of the two subgroups A and B. Each dataset contains a sample of 10,000 data points, with samples distributed according the values of $c_A$ and $c_B$, parameters which control the proportion of the data coming from each subgroup. The accuracy of the models on the three datasets is summarized in Tables 5 and 6 below. For both ERM and DRO we use L2-regularized logistic regression trained with stochastic gradient descent. The step-size for all algorithms is fixed at 0.05 and we train for 15,000 epochs.

Models trained with the distributionally robust objective have the additional complication that we must specify a value for the radius of the $\chi^2$-divergence ball, *i.e.* $\rho$. The larger the value of $\rho$, the more we can expect a DRO model to differ from an ERM model because as the $\chi^2$-divergence ball grows, the worst case distribution can be further and further from the data generating distribution. Conversely, in the limit as $\rho \to 0$, we recover ERM as the $\chi^2$-divergence ball shrinks to contain only the data generating distribution.

Choosing the value of $\rho$ is a challenging decision, as the performance of a model varies significantly as $\rho$ changes. If one has access to demographic information, it is possible to conduct a grid search over possible $\rho$ values in order to find a value that results in a model with the desirable fairness properties. Doing this, however, largely defeats the purpose of DRO. As explained earlier, a central advantage to using DRO, rather than some fairness constrained optimization technique, is that DRO does not require access to demographic information.

In this experiment we work directly with the dual formulation of DRO and set $\eta = 0.56$. The dual formulation of DRO is explained in A.2 This value was chosen empirically to achieve relatively uniform accuracy across group A and group B for an 80/20 split between the two subgroups. As the values of $c_A$ and $c_B$ change, we can see that the performance of DRO changes for a given value of $\eta$ and hence $\rho$, as $\eta^*$ depends on $\rho$.

| Objective | Group | $[c_A = 0.6, c_B = 0.4]$ | $[c_A = 0.8, c_B = 0.2]$ | $[c_A = 0.95, c_B = 0.05]$ |
|---|---|---|---|---|
| ERM | A | 0.797 | 0.907 | 0.966 |
| | B | 0.701 | 0.652 | 0.592 |
| | All Data | 0.759 | 0.856 | 0.948 |
| DRO | A | 0.665 | 0.751 | 0.780 |
| | B | 0.869 | 0.744 | 0.766 |
| | All Data | 0.747 | 0.750 | 0.780 |

Table 1: Accuracy by Group for ERM and DRO.

We first examine the performance of ERM (Table 1). As the data is not linearly separable, ERM must learn a decision boundary that trades off performance between the two subgroups. Because the ERM objective treats the loss on each data point equally, the model learns a decision boundary that is more accurate for the majority group than for the minority group. This discrepancy in accuracy of predictions worsens the smaller the majority group is. For instance, when 95% of the data comes from group A, the logistic regression model trained with an ERM objective achieves 96% accuracy on group A members, but only 59.2% accuracy on group B members.

Models trained with a DRO objective behave much differently (Table 1). For the 80/20 split and 95/5 split, the DRO models learn relatively fair decision boundaries, effectively balancing perfor-

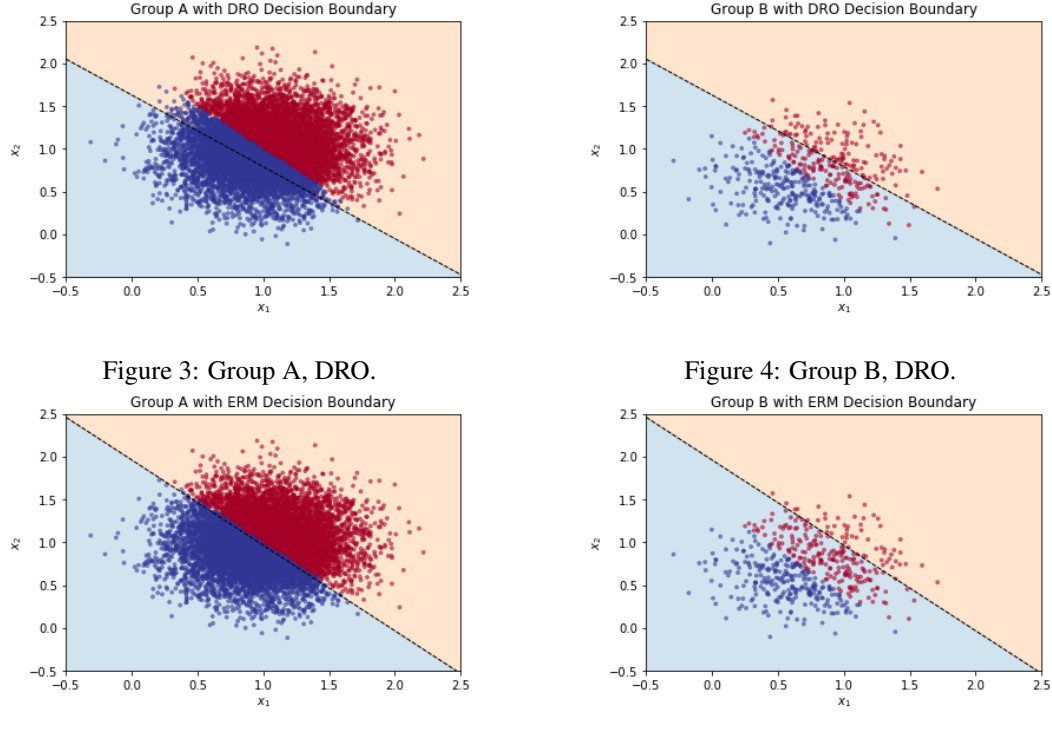

Figure 3: Group A, DRO.

Figure 4: Group B, DRO.

Figure 5: Group A, ERM.

Figure 6: Group B, ERM.

Figure 7: Decision boundaries for ERM and DRO classifiers with samples from groups A and B with $c_A = 0.95$ and $c_B = 0.05$. Shading indicates predicted label and data point colour indicates true label.

mance on both subgroups A and B. For the 60/40 split, however, DRO actually learns a model that performs significantly better on the minority group than the majority group. This model is in a sense discriminatory against the majority group. This is the result of the radius of the $\chi^2$-divergence ball being too large and the model thus overly focusing on a worst case distribution.

In Figures 3, 4, 5, and 6 we plot data from groups A and B for data generated with $c_A = 0.95$, $c_B = 0.05$ with the learned decision boundary from a DRO and ERM model, respectively, overlaid. The background shading indicates the predicted label with blue representing $\hat{y} = 0$ and red representing $\hat{y} = 1$, while the colour of the data points indicate their true label.

## 4.3 RRM AND RDRO WITH SUBGROUPS

We have seen empirical confirmation of convergence of RDRO and RRM in our first experiment, and the static classification with subgroups has demonstrated the potential for DRO as a tool to achieve fairness in ML models, but neither of these address the question of long term fairness under dynamically changing distributions. To investigate this question, we generate data in a similar fashion to the static classification experiment (details in A.6), and add a performative element to the model.

The data in this experiment is composed of a mixture of multivariate normal distributions with different means. 80% of the data is drawn from the majority subgroup, while the remaining 20% is drawn from the minority subgroup. We again implement a strategic manipulation distribution map and train models using RRM and RDRO for 30 iterations. As we saw with the *Give Me Some Credit* dataset, for small values of $\epsilon$ we observe convergence, whereas for larger values of $\epsilon$ we do not observe convergence. Plots illustrating this behaviour can be found in A.6.

The accuracy of the models for the values of $\epsilon$ for which we observe convergence are given in Tables 4.3 and 4.3. We can see that RRM converges to a performatively stable model that achieves high

| ERM Performative Accuracy | | | | DRO Performative Accuracy | | |
| --- | --- | --- | --- | --- | --- | --- |
| Group | $\epsilon = 0.01$ | $\epsilon = 0.25$ | $\epsilon = 0.5$ | $\epsilon = 0.01$ | $\epsilon = 0.25$ | $\epsilon = 0.5$ |
| A | 0.893 | 0.896 | 0.898 | 0.687 | 0.710 | 0.738 |
| B | 0.540 | 0.540 | 0.540 | 0.670 | 0.660 | 0.660 |
| All Data | 0.834 | 0.837 | 0.838 | 0.684 | 0.701 | 0.725 |

Table 2: Accuracy by Group for ERM and DRO after 30 iterations.

accuracy on the majority group, but performs poorly on the minority group. RDRO on the other hand, results in models that perform similarly across both subgroups in the data.

This result, while perhaps not that surprising in light of the two previous experiments, is important, as it demonstrates that not only does DRO exhibit similar convergence behaviour to ERM, but DRO converges to fair fixed points, whereas ERM converges to discriminatory fixed points in the presence of heterogeneous data composed of minority and majority subgroups. Recall also that DRO is not given access to group information, but still learns to achieve more uniform performance across subgroups, as it is attempting to minimize the worst case loss across all probability distributions within the $\chi^2$-divergence ball surrounding the data generating distribution. Further details of this experiment, as well as additonal experiments are provided in A.6.

## 5 CONCLUSION AND FUTURE WORK

Performative prediction offers a rigorous framework in which to reason about decision making in dynamic environments. Existing theory for performative prediction had only been developed for models trained with ERM objectives, however, which is liable to result in models that discriminate against minority groups.

We extend the performative prediction framework to DRO and prove a convergence result for RDRO. We also investigate RRM and RDRO empirically and find that while they exhibit similar convergence behaviour, RRM converges to fixed points that perform well on a majority subgroup, but poorly on a minority subgroup, whereas RDRO converges to models which successfully balance performance across majority and minority subgroups.

We hope that this work helps to further our still very limited understanding of fairness properties of ML algorithms in changing and dynamic environments, which charactierize many real-world settings with fairness concerns. By combining performative prediction with a distributionally robust objective, we have tried to overcome some of the shortcomings of the dominant fairness criteria and to offer a flexible and general tool to ML practitioners and researchers concerned with ensuring that algorithms do not discriminate against vulnerable minority groups.

This work represents a preliminary investigation into the combination of DRO and performative prediction as a method to understand and learn fair algorithms, and, as such, leaves many important avenues for future work open. Further theoretical investigation of the convergence properties of RDRO would yield better understanding of the distributionally robust objective, as this work eschewed answering important questions of stability of worst case distributions as well as questions of smoothness and strong convexity of distributionally robust objectives.

It would also be very interesting to investigate both the convergence properties and practical efficacy of a parametric DRO objective, rather than an $f$-divergence based robust objective. Recent work suggests that parametric DRO can outperform $f$-divergence based DRO and offers practitioners and theoreticians greater flexibility than with analytic DRO Michel et al. (2022; 2021).

Additionally, performative prediction and distributionally robust optimization have connections and similarities to many other well established areas of research in machine learning, and it is likely that existing results in areas such as bandits, reinforcement learning, online learning, stochastic optimization, and out of distribution generalization could greatly contribute to our understanding of DRO and performative prediction.

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

## A APPENDIX

### A.1 RELATED WORK

The two most influential areas of research for this work are performative prediction and distributionally robust optimization, and the papers most important for this work are Perdomo et al. (2020) and Duchi & Namkoong (2021). Perdomo *et al.* develop the performative prediction framework, while Duchi and Namkoong give a thorough summary of distributionally robust optimization and explain its potential application to fairness concerns in machine learning.

Following the publication of *Performative Prediction* Perdomo et al. (2020), there have been a number of papers which have further explored performative prediction and extended some of the results from the original paper. For instance, Mendler-Dünner et al. (2020) prove results for stochastic optimization in performative prediction, Miller et al. (2021) prove new results relating performatively stable points to performatively optimal points, Brown et al. (2020) attempt to move toward sequential games for performative prediction by adding a notion of state to the performative prediction problem, and Dong & Ratliff (2021) approach performative prediction from a dynamical systems perspective allowing a move away from strict contraction mappings when examining convergence to performatively stable models.

Distributionally robust optimization is an older area of research as compared to performative prediction, but it has only recently received more attention within the machine learning community, largely due to work by Hongseook Namkoong and John Duchi Namkoong & Duchi (2017); Duchi et al. (2020); Jeong & Namkoong (2020). Areas such as finance, where robust optimization is important as rare events have the potential to be catastrophic for a portfolio, have seen research on robust optimization for decades. See Ben-Tal et al. (2009) for a survey. Ben-Tal et al. (2013) Shapiro (2017) provide rigorous mathematical treatments of DRO with $f$-divergence balls.

A number of recent papers have also attempted to better understand fairness in machine learning over time, albeit not within the performative prediction framework. Closely related to performative prediction, strategic classification Hardt et al. (2016a) models the prediction problem as an adversarial game where individuals manipulate their features in response to deployed models. Recent work Milli et al. (2019); Hu et al. (2019) has examined how strategic classification interacts with fairness concerns. Strategic classification is an interesting problem setting, but is not as general as performative prediction and only captures scenarios that can rightly be modelled as adversarial games. D'Amour et al. (2020) on the other hand, approach long-term fairness questions from a purely empirical perspective and use simulation studies to understand the long-term dynamics of algorithmic choices on populations in a variety of scenarios designed to reflect real-world applications. This approach is interesting, but the lack of theoretical framework limits their ability to generalize beyond specific simulations.

Finally, the work most closely related to this paper, Hashimoto et al. (2018), examines the impact of repeatedly minimizing loss and deploying a model using empirical risk minimization and distributionally robust optimization on a population that changes as a function of the loss incurred. This work, however, does not fit into the performative prediction framework and applies only to a specific scenario in which individuals arrive according to a Poisson process and depart as a function of the loss. This research aims to extend this work by combining DRO and performative prediction in order to leverage the known results in both areas to provide a more general understanding of how DRO can impact long-term fairness.

### A.2 DISTRIBUTIONALLY ROBUST OPTIMIZATION

The notation $Q \ll P_0$ means that $Q$ is absolutely continuous with respect to $P_0{}^2$. An $f$-divergence, $D_f(Q||P)$, is a function that measures the difference between two probability distributions $Q$ and $P$, although it is not a metric. There are a variety of different $f$-divergences, but their general definition is as follows Renyi (1961); Csiszár & Shields (2004); Csiszar (1967).

---

[2]If two measures, $\mu$ and $\nu$, are on the same measure space $(\mathcal{X}, \mathcal{A})$, $\mu$ is said to be absolutely continuous with respect to $\nu$ if $\mu(A) = 0$ for every set $A$ for which $\nu(A) = 0$.

$D_f(Q||P) := \int f(\frac{dQ}{dP})dP$ where $f$ is a convex function such that $f(1) = 0$. If $Q$ and $P$ are absolutely continuous with respect to a reference distribution $\mu$, then their probability densities $q$ and $p$ satisfy $dP = p$ and $dQ = q$ and the $f$-divergence can be written as

$$D_f(Q||P) = \int f(\frac{q(x)}{p(x)})d\mu(x).$$

Some examples of $f$-divergences are KL-divergence ($f(t) = t\log t$), Pearson $\chi^2$-divergence ($f(t) = (t-1)^2, t^2 - 1, t^2 - t$), and total variation distance ($f(t) = \frac{1}{2}|t-1|$). An $f$-divergence ball centred at $P$ of radius $\rho$ is a set that contains all probability distributions whose $f$-divergence is within $\rho$ of $P$. For example, $\{Q : D_f(Q||P_0) \leq \rho\}$ is an $f$-divergence ball that contains all probability distributions $Q$ that are within $\rho$ $f$-divergence distance from $P_0$. This is analogous to the notion of an $\epsilon$-ball surrounding a vector in a vector space.

The distributionally robust problem therefore, is to find a set of parameters, $\theta \in \Theta$, that minimize the worst case expected loss of all probability distributions $Q$ that are within the $f$-divergence ball of radius $\rho$ of our data generating distribution. This differs from ERM in that we do not seek to minimize the expected loss over our data generating distribution, but instead seek to minimize the worst case expected loss over a set of probability distributions nearby our data generating distribution.

As the name suggests, learning parameters that minimize the distributionally robust risk gives us a model that is robust to changes in the data generating distribution. Traditional risk minimization will return a model that minimizes risk for the data generating distribution, but offers no performance guarantees when that distribution changes, and can result in models that are brittle and do not perform well on out of distribution (OOD) examples. DRO, on the other hand, is a more conservative procedure that minimizes worst-case risk and should thus be robust to changes to the probability generating distribution that lie within the $f$-divergence ball of radius $\rho$.

As with traditional risk minimization, we do not have access to the theoretical data generating distribution. Instead, we solve the distributionally robust problem via the plug-in estimator

$$\hat{\theta}_n \in \arg\min_{\theta \in \Theta} \left\{ R_f(\theta, \hat{P}_n) := \sup_{Q \ll \hat{P}_n} \{\mathbb{E}_Q[\ell(\theta; X)] : D_f(Q||\hat{P}_n) \leq \rho\} \right\}$$

where $\hat{P}_n$ is the empirical measure on $X_i \sim^{iid} P_0$. Duchi and Namkoong prove convergence guarantees and rates for the plug-in estimator along with some asymptotic results such as showing consistency of the estimator Duchi & Namkoong (2021).

It is not necessarily clear from the discussion so far what DRO has to do with fairness concerns in machine learning, but DRO in fact has characteristics that are very desirable for addressing issues of fairness and discrimination in learning algorithms. We will explain this shortly, but first we introduce dual reformulations of the distributionally robust problem as it provides some intuition as to the connection with fairness concerns.

The following theorem comes from Shapiro (2017). Let $f^*(s) := \sup_t\{st - f(t)\}$ be the Fenchel conjugate.

**Theorem A.1.** *([Shapiro (2017), Section 3.2]). Let $P$ be a probability measure on $(\mathcal{X}, \mathcal{A})$ and $\rho > 0$. Then*

$$R_f(\theta; P) = \inf_{\lambda \geq 0, \eta \in \mathbb{R}} \left\{ \mathbb{E}_P\left[\lambda f^*\left(\frac{\ell(\theta; X) - \eta}{\lambda}\right)\right] + \lambda\rho + \eta \right\}$$

*for all $\theta$. Moreover, if the supremum on the left hand side is finite, there are finite $\lambda(\theta) \geq 0$ and $\eta(\theta) \in \mathbb{R}$ attaining the infimum on the right hand side.*

Additionally, Duchi & Namkoong (2021) provide a simplified version of this dual formulation for the Cressie-Read family of $f$-divergences, obtained by minimizing out $\lambda > 0$ from Theorem A.1.

**Theorem A.2.** *([Duchi & Namkoong (2021), Section 2]). For any probability $P$ on $(\mathcal{X}, \mathcal{A})$, $k \in (1, \infty)$, $k_* = k/(k-1)$, any $\rho > 0$, and $c_k(\rho) = (1 + k(k-1)\rho)^{1/k}$, we have for all $\theta \in \Theta$*

$$R_f(\theta; P) = \inf_{\eta \in \mathbb{R}} \left\{ c_k(\rho)\mathbb{E}_P\left[(\ell(\theta; X) - \eta)_+^{k_*}\right]^{1/k_*} + \eta \right\}$$

The above formulations are jointly convex in $(\theta, \lambda, \eta)$ and $(\theta, \eta)$, respectively, for convex losses, $\ell(\theta; X)$, making them amenable to techniques from convex optimization such as interior point methods Boyd & Vandenberghe (2004).

The Cressie-Read family of $f$-divergences are parameterized by $k \in (-\infty, \infty) \setminus \{0, 1\}$, $k_* = \frac{k}{k-1}$, with

$$f_k(t) := \frac{t^k - kt + k - 1}{k(k-1)} \quad \text{and} \quad f_k^*(s) := \frac{1}{k}\left[((k-1)s+1)_+^{k_*} - 1\right].$$

Issues of fairness arise in machine learning when an algorithm treats different demographic groups in a disparate fashion. As outlined above, this often means that algorithmic performance varies across different demographic groups. If this occurs, it is almost certainly the case that there exist distinct probability distributions over the demographic groups because if all demographic groups were characterized the the same probability distribution, classification algorithms should exhibit uniform performance across demographic groups.

This realization makes it easy to see why ERM is likely to discriminate, particularly against minority groups. ERM treats the loss on each data point equally, thus, if there is a majority and minority probability distribution and a learning algorithm must balance performance on these groups, an ERM objective is likely to result in a model that performs better on the majority group and worse on the minority group. It is exactly this type of problem that DRO is designed to resolve.

Unlike ERM, DRO does not equally weight each data point, but instead up-weights data points on which the model is achieving high loss. This means that the model should achieve somewhat uniform performance on individuals across demographic groups. This can be seen by looking at the dual formulation in Theorem A.2. The DRO objective only considers losses above the optimal dual variable $\eta^*(\theta)$ and these losses are up-weighted by the $L^{k_*}(P)$-norm. Losses that are less than the optimal dual variable are set to zero in the objective. Another way of saying this is that the DRO objective is equivalent to optimizing the tail-performance of a model Duchi & Namkoong (2021).

If DRO performs poorly on a subset of the data that is correlated with a distinct demographic group, these losses will be up-weighted by the dual formulation of the distributionally robust objective, pushing the model to improve performance on this subset. DRO does not offer any guarantees of uniform performance across demographic groups, but as we increase the value of $\rho$ we will increase the loss incurred on "hard" regions of the data where the model performs poorly.

### A.3   Performative Prediction

In supervised learning we assume that our data is sampled i.i.d. from an unknown data generating distribution and that our model is then deployed to make predictions on data that follows this same distribution. In many scenarios, however, the very act of making predictions influences the data on which we wish to make these predictions. That is to say, our models are *performative* and instead of passively describing the world and making predictions about it, they actually induce change in the world.

A simple example of this is predictive policing. In predictive policing we train a model to predict where crimes are likely to occur based on historical data and then deploy more resources to areas where the model predicts crimes are more likely to occur. The increased police patrols and surveillance results in more crimes being detected which might further increase the perceived crime rate in those areas. If this data is then used for future predictions it will result in a shifted distribution of the data as a result of the predictions of the previous model. Another example of performativity is an algorithm that weights different elements of a student's CV such as SAT scores and GPA in order to make college admissions decisions. If the algorithm heavily weights SAT scores, over time it will become apparent to applicants that SAT scores are very important and they will dedicate more resources to improving SAT scores, thus changing the distribution of the data on which the algorithm is making predictions as a result of those very predictions.

Performative prediction is closely related to many other fields in machine learning including bandits, reinforcement learning, strategic classification, causal inference, convex optimization, and game theory, but the precise notion and formalism for performative prediction was only developed very

recently in Perdomo et al. (2020). We formally specify the performative prediction problem now and contrast it with the supervised learning problem.

Assume we have a measure space $(\mathcal{Z}, \mathcal{A})$ with $Z$ a random element of $\mathcal{Z}$ and $\mathcal{D}$ the data generating distribution on this space. Let $\Theta \subset \mathbb{R}^d$ be the parameter (model) space and $\ell : \Theta \times \mathcal{Z} \to \mathbb{R}$ be a loss function. The supervised learning problem is to minimize the the objective over this distribution. In the case of risk minimization the objective is the expected loss, $\ell(Z; \theta)$ with respect to $\mathcal{D}$.

$$R(\theta) = \mathbb{E}_{Z \sim \mathcal{D}}[\ell(Z; \theta)].$$

In contrast to this, performative prediction involves making predictions on a distribution that has been shifted as a result of deploying the model, $\mathcal{D}(\theta)$. We refer to $\mathcal{D}(\theta)$ as the *distribution map*. The concept that captures this notion of risk is known as *performative risk* and is formalized as follows:

$$PR(\theta) = \mathbb{E}_{Z \sim \mathcal{D}(\theta)}[\ell(Z; \theta)].$$

The difference between this and the supervised learning problem is that expected loss is now taken with respect to the induced distribution rather than the data generating distribution.

The notion of what constitutes a good model is different in supervised learning and performative prediction. In supervised learning the task is simpler - minimize the risk on the data generating distribution. In performative prediction however, we now need to consider how to minimize risk on a distribution that is different from that which generated our training data, and is in fact a function of whatever model we deploy. To capture these notions, Perdomo et al. (2020) define *performative optimality* and *performative stability*.

**Definition A.3.** (performative optimality) A model $f_{\theta_{PO}}$ is performatively optimal if the following relationship holds:

$$\theta_{PO} = \arg \min_{\theta \in \Theta} \mathbb{E}_{Z \sim \mathcal{D}(\theta)}[\ell(Z; \theta)].$$

Equivalently, $\theta_{PO} = \arg \min_{\theta \in \Theta} PR(\theta)$ where $PR(\theta)$ is the performative risk as defined above.

A performatively optimal point is a minimizer of the performative risk. An alternative solution concept is referred to as *performative stability*.

**Definition A.4.** (performative stability) A model $f_{\theta_{PS}}$ is performatively stable if the following relationship holds:

$$\theta_{PS} = \arg \min_{\theta \in \Theta} \mathbb{E}_{Z \sim \mathcal{D}(\theta_{PS})}[\ell(Z; \theta)].$$

Define $DPR(\theta, \theta') := \mathbb{E}_{Z \sim \mathcal{D}(\theta)}[\ell(Z; \theta')]$ as the decoupled performative risk; then $\theta_{PS} = \arg \min_{\theta \in \Theta} DPR(\theta_{PS}, \theta)$.

A performatively stable model is not necessarily a minimizer of the performative risk, but it is optimal on the distribution it induces. Hence, if you have performative stability there is no need to retrain a model to cope with the induced distribution drift. Performative optimality and performative stability are distinct concepts and a performatively optimal point is not necessarily performatively stable, and vice versa. As explained in Perdomo et al. (2020), performatively stable models are fixed points of risk minimization. In game theoretic terms, we can consider performative prediction as a game in which one player deploys a model, $\theta$, and the environment responds with some distribution map, $\mathcal{D}(\theta)$. If $\mathcal{D}(\theta)$ is a best response, then a performatively optimal point corresponds to a Stackelberg equilibrium, whereas a performatively stable point corresponds to a Nash equilibrium. From game theory we know that except in special cases (*e.g.* finite zero-sum games), Nash equilibria and Stackelberg equilibria do not necessarily coincide Kroer (2022).

Performative prediction presents a special case of learning under distribution drift, where the distribution drift is a function of the model deployed. A common approach in supervised learning under distribution drift is to retrain a model on newly collected data. While this does not directly minimize performative risk, it is a potentially reasonable solution in a variety of scenarios. Perdomo et al. (2020) prove theorems under which repeated risk minimization and repeated gradient descent converge to performatively stable models. We present these findings in detail here as they are relevant for our work.

**Definition A.5.** ($\epsilon$-sensitivity) We say that a distribution map $\mathcal{D}(\cdot)$ is $\epsilon$-sensitive if for all $\theta, \theta' \in \Theta$:

$$W_1\left(\mathcal{D}(\theta), \mathcal{D}(\theta')\right) \leq \epsilon ||\theta - \theta'||_2,$$

where $W_1$ denotes the Wasserstein-1 distance.

We also make assumptions on the loss function $\ell(z; \theta)$. Let $\mathcal{Z} := \cup_{\theta \in \Theta} supp(\mathcal{D}(\theta))$.

**Definition A.6.** (joint smoothness) We say that a loss function $\ell(z; \theta)$ is $\beta$-jointly smooth if the gradient $\nabla_\theta$ is $\beta$-Lipschitz in $\theta$ and $z$, that is

$$||\nabla_\theta \ell(z; \theta) - \nabla_\theta \ell(z; \theta')||_2 \leq \beta ||\theta - \theta'||_2, \quad ||\nabla_\theta \ell(z; \theta) - \nabla_\theta \ell(z'; \theta)||_2 \leq \beta ||z - z'||_2,$$

for all $\theta, \theta' \in \Theta$ and $z, z' \in \mathcal{Z}$

**Definition A.7.** (strong convexity) We say that a loss function $\ell(z; \theta)$ is $\gamma$-strongly convex if

$$\ell(z; \theta) \geq \ell(z; \theta') + \nabla_\theta \ell(z; \theta')^T (\theta - \theta') + \frac{\gamma}{2} ||\theta - \theta'||_2^2,$$

for all $\theta, \theta' \in \Theta$ and $z \in \mathcal{Z}$. If $\gamma = 0$ this condition is equivalent to convexity.

**Definition A.8.** (repeated risk minimization) Repeated risk minimization (RRM) refers to the procedure where, starting from an initial model $f_{\theta_0}$, we perform the following sequence of updates for every $t \geq 0$:

$$\theta_{t+1} = G(\theta_t) := \arg\min_{\theta \in \Theta} \mathbb{E}_{Z \sim \mathcal{D}(\theta_t)} [\ell(Z; \theta)].$$

Definitions A.5, A.6, and A.7 are assumptions on the loss and distribution map that are required for Theorem 3.5 in Perdomo et al. (2020). We state that theorem now.

**Theorem A.9.** *([Perdomo et al. (2020), Theorem 3.5]) Suppose that the loss $\ell(z; \theta)$ is $\beta$-jointly smooth and $\gamma$-strongly convex. If the distribution map $\mathcal{D}(\cdot)$ is $\epsilon$-sensitive, then the following statements are true:*

1. *$||G(\theta) - G(\theta')||_2 \leq \epsilon \frac{\beta}{\gamma} ||\theta - \theta'||_2$, for all $\theta, \theta' \in \Theta$*

2. *If $\epsilon < \frac{\gamma}{\beta}$, the iterates $\theta_t$ of RRM converge to a uniquely performatively stable point $\theta_{PS}$ at a linear rate: $||\theta_t - \theta_{PS}||_2 \leq \delta$ for $t \geq \left(1 - \epsilon \frac{\beta}{\gamma}\right)^{-1} \log\left(\frac{||\theta_0 - \theta_{PS}||_2}{\delta}\right)$*

This theorem says that with the appropriate smoothness and convexity conditions satisfied on the loss and distribution map, repeatedly retraining a model by minimizing risk will converge to a unique performatively stable model at a linear rate. The proof proceeds by demonstrating that under the assumptions stated in Theorem A.9, the RRM operator is a contraction mapping. Perdomo et al. (2020) also show that without any of $\epsilon$-sensitivty, $\beta$-joint smoothness, or $\gamma$-strong convexity one can produce examples that do not converge to a fixed point. Hence these assumptions are necessary conditions to guarantee convergence of RRM to a performatively stable model in the general case.

## A.4 DEFINITIONS AND PROOF FOR DISTRIBUTIONALLY ROBUST PERFORMATIVE PREDICTION

We begin by redefining performative risk and performatively optimal and stable models in terms of a robust objective. In the following $\mathcal{D}(\theta)$ is a distribution map and $D_f(Q||\mathcal{D}(\theta))$ is an $f$-divergence ball of radius $\rho$.

**Definition A.10.** (robust performative risk) The robust performative risk of a model, $\theta$, is

$$RPR(\theta) = \sup_{Q \ll \mathcal{D}(\theta)} \{\mathbb{E}_{Z \sim Q}[\ell(Z; \theta)] : D_f(Q||\mathcal{D}(\theta)) \leq \rho\}.$$

Robust performative risk differs from performative risk in that the induced distribution, $\mathcal{D}(\theta)$, is now the centre of an $f$-divergence ball over which a supremum of the expected loss is taken.

**Definition A.11.** (robust performative optimality) A model $f_{\theta_{PO}}$ is robustly performatively optimal if the following relationship holds:

$$\theta_{PO} = \arg\min_{\theta \in \Theta} \sup_{Q \ll \mathcal{D}(\theta)} \{\mathbb{E}_{Z \sim Q}[\ell(Z; \theta)] : D_f(Q||\mathcal{D}(\theta)) \leq \rho\}.$$

Equivalently, $\theta_{PO} = \arg\min_{\theta \in \Theta} RPR(\theta)$ where $RPR(\theta)$ is the robust performative risk as defined above.

**Definition A.12.** (robust performative stability) A model $f_{\theta_{PS}}$ is robustly performatively stable if the following relationship holds:

$$\theta_{PS} = \underset{\theta \in \Theta}{\arg\min} \sup_{Q \ll \mathcal{D}(\theta_{PS})} \{\mathbb{E}_{Z \sim Q}[\ell(Z;\theta)] : D_f(Q||\mathcal{D}(\theta_{PS})) \leq \rho\}.$$

Define $RDPR(\theta, \theta') := \sup_{Q \ll \mathcal{D}(\theta)} \{\mathbb{E}_{Z \sim Q}[\ell(Z;\theta')] : D_f(Q||\mathcal{D}(\theta)) \leq \rho\}$ as the robust decoupled performative risk; then $\theta_{PS} = \arg\min_{\theta \in \Theta} RDPR(\theta_{PS}, \theta)$.

We discussed earlier that previous work has shown that repeated risk minimization converges to a performatively stable model under certain assumptions on the loss and distribution map Perdomo et al. (2020). We can analogously define repeated distributionally robust optimization as follows.

**Definition A.13.** (repeated distributionally robust optimization) Repeated distributionally robust optimization (RDRO) refers to the procedure where, starting from an initial model $f_{\theta_0}$, we perform the following sequence of updates for every $t \geq 0$:

$$\theta_{t+1} = G(\theta_t) := \underset{\theta \in \Theta}{\arg\min} \sup_{Q \ll \mathcal{D}(\theta_t)} \{\mathbb{E}_{Z \sim Q}[\ell(Z;\theta)] : D_f(Q||\mathcal{D}(\theta_t)) \leq \rho\}.$$

As with RRM, this is an iterative procedure where we optimize the distributionally robust objective at each time step $t$. While it is only a small departure from RRM conceptually, the DRO objective has fundamentally different mathematical properties than risk minimization making it unclear whether RDRO and RRM should exhibit similar behaviour.

We now re-state and prove our convergence theorem for RDRO. The proof of this theorem is a straightforward adaptation of the proof in Perdomo et al. (2020). We first introduce two lemmas used in Perdomo et al. (2020) which we will make use of in our proof.

**Lemma A.14.** *(First-order optimality condition) Let $f$ be convex and let $\Omega$ be a closed convex set on which $f$ is differentiable, then*

$$x_* \in \underset{x \in \Omega}{\arg\min} f(x)$$

*if and only if*

$$\nabla f(x_*)^T(y - x_*) \geq 0, \forall y \in \Omega.$$

**Lemma A.15.** *(Kantorovich-Rubinstein) A distribution map $\mathcal{D}(\cdot)$ is robustly $\epsilon$-sensitive if and only if for all $\theta, \theta' \in \Theta$:*

$$\sup\left\{\left|\mathbb{E}_{Z \sim \mathcal{D}^*(\theta)}[g(Z)] - \mathbb{E}_{Z \sim \mathcal{D}^*(\theta')}[g(Z)]\right| : g : \mathbb{R}^P \to \mathbb{R}, g \text{ 1-Lipschitz}\right\} \leq \omega\epsilon||\theta - \theta'||_2$$

*where $\mathcal{D}^*(\theta) = \underset{Q:D_f(Q||\mathcal{D}(\theta)) \leq \rho}{\arg\max} \mathbb{E}_{Z \sim Q}[\ell(Z;\theta)]$.*

We now state our theorem.

**Theorem A.16.** *Suppose that the distributionally robust objective satisfies definitions 3.5 and 3.6, and that the distribution map satisfies definition 3.7. Then the following statements are true:*

1. *$||G(\theta) - G(\theta')||_2 \leq \omega\epsilon\frac{\beta}{\gamma}||\theta - \theta'||_2$, for all $\theta, \theta' \in \Theta$*

2. *If $\omega\epsilon < \frac{\gamma}{\beta}$, the iterates $\theta_t$ of RDRO converge to a unique robustly performatively stable point $\theta_{PS}$ at a linear rate: $||\theta_t - \theta_{PS}||_2 \leq \delta$ for $t \geq \left(1 - \omega\epsilon\frac{\beta}{\gamma}\right)^{-1} \log\left(\frac{||\theta_0 - \theta_{PS}||_2}{\delta}\right)$.*

*Note that $G(\theta) := \arg\min_{\theta \in \Theta} \sup_{Q \ll \mathcal{D}(\theta)} \{\mathbb{E}_{Z \sim Q}[\ell(Z;\theta)] : D_f(Q||\mathcal{D}(\theta)) \leq \rho\}$.*

*Proof.* Fix $\theta, \theta' \in \Theta$. Let

$$\mathcal{D}^*(\theta) = \underset{Q:D_f(Q||\mathcal{D}(\theta)) \leq \rho}{\arg\max} \mathbb{E}_{Z \sim Q}[\ell(Z;\theta)].$$

That is, $\mathcal{D}^*(\theta)$ is the distribution within the $f$-divergence ball centred at $\mathcal{D}(\theta)$ with radius $\rho$ that maximizes the expected loss. Further, let

$$f(\xi) = \mathbb{E}_{Z \sim \mathcal{D}^*(\theta)}[\ell(Z;\xi)] \quad \text{and} \quad f'(\xi) = \mathbb{E}_{Z \sim \mathcal{D}^*(\theta')}[\ell(Z;\xi)].$$

That is, $f(\xi)$ and $f'(\xi)$ are the worst case losses for $f$-divergence balls centred at $\theta$ and $\theta'$ respectively with radius $\rho$.

We now use Definition 3.6 and Definition 3.5 to ensure that $f$ is $\gamma$-strongly convex and that the gradient of $f$ exists and is $\beta$-jointly smooth. With our assumption of $\gamma$-strong convexity of $f$, $G(\theta)$ is the unique minimizer of $f(x)$ and we have the following two inequalities

$$f(G(\theta)) - f(G(\theta')) \geq (G(\theta) - G(\theta'))^T \nabla f(G(\theta')) + \frac{\gamma}{2}||G(\theta) - G(\theta')||_2^2 \tag{1}$$

$$f(G(\theta')) - f(G(\theta)) \geq \frac{\gamma}{2}||G(\theta) - G(\theta')||_2^2 \tag{2}$$

Inequality (1) comes from the definition of $\gamma$-strong convexity and inequality (2) comes from $\gamma$-strong convexity and the first-order optimality condition since Lemma A.14 tells us $(G(\theta') - G(\theta))^T \nabla f(G(\theta)) \geq 0$ because $G(\theta)$ is the minimizer of $f(x)$.

Using these two inequalities we can derive the following

$$-\gamma||G(\theta) - G(\theta')||_2^2 \geq f(G(\theta)) - f(G(\theta')) - \frac{\gamma}{2}||G(\theta) - G(\theta')||_2^2$$
$$\geq (G(\theta) - G(\theta'))^T \nabla f(G(\theta'))$$

where we get the first inequality from (2) and the second from (1).

Now, using $\beta$-joint smoothness in $z$ and Cauchy-Schwarz we get

$$||(G(\theta) - G(\theta'))^T \nabla_\theta \ell(z; G(\theta')) - (G(\theta) - G(\theta'))^T \nabla_\theta \ell(z'; G(\theta'))||_2$$
$$\leq ||G(\theta) - G(\theta')||_2 \beta ||z - z'||_2$$

That is, $(G(\theta) - G(\theta'))^T \nabla_\theta \ell(z; G(\theta'))$ is $||G(\theta) - G(\theta')||_2 \beta$-Lipschitz in $z$. We will now use this and Kantorovich-Rubinstein (Lemma A.15). Let

$$g(z) = \frac{(G(\theta) - G(\theta'))^T \nabla_\theta \ell(z; G(\theta'))}{||G(\theta) - G(\theta')||_2 \beta}$$

The function $g(z)$ is 1-Lipschitz in $z$ because we have just divided $(G(\theta) - G(\theta'))^T \nabla_\theta \ell(z; G(\theta'))$ by its Lipschitz constant. From Lemma A.15 and Definition 3.7 we have the following, with $g(Z)$ as defined above

$$\mathbb{E}_{Z \sim \mathcal{D}^*(\theta)}[g(Z)] - \mathbb{E}_{Z \sim \mathcal{D}^*(\theta')}[g(Z)] \leq \omega\epsilon||\theta - \theta'||_2$$

Using linearity of expectation and multiplying by $||G(\theta) - G(\theta')||_2 \beta$ we get the following

$$(G(\theta) - G(\theta'))^T \nabla f(G(\theta')) - (G(\theta) - G(\theta'))^T \nabla f'(G(\theta'))$$
$$\geq -\omega\epsilon\beta||G(\theta) - G(\theta')||_2||\theta - \theta'||_2$$

Now again using Lemma A.14, we have $(G(\theta) - G(\theta'))^T \nabla f'(G(\theta')) \geq 0$, hence $(G(\theta) - G(\theta'))^T \nabla f(G(\theta')) \geq \omega\epsilon\beta||G(\theta) - G(\theta')||_2||\theta - \theta'||_2$. From our work above we showed that $-\gamma||G(\theta) - G(\theta')||_2^2 \geq (G(\theta) - G(\theta'))^T \nabla f(G(\theta'))$. Putting this all together we get

$$-\gamma||G(\theta) - G(\theta')||_2^2 \geq -\omega\epsilon\beta||G(\theta) - G(\theta')||_2||\theta - \theta'||_2$$

We rearrange the above to get

$$||G(\theta) - G(\theta')||_2 \leq \omega\epsilon\frac{\beta}{\gamma}||\theta - \theta'||_2$$

which proves claim (1) of the theorem.

Claim (2) follows easily. We note that $\theta_t = G(\theta_{t-1})$ from the definition of RDRO and $G(\theta_{PS}) = \theta_{PS}$ by the definition of robust performative stability. Using the result of claim (1) we get

$$||\theta_t - \theta_{PS}||_2 \leq \omega\epsilon\frac{\beta}{\gamma}||\theta_{t-1} - \theta_{PS}||_2 \leq \left(\omega\epsilon\frac{\beta}{\gamma}\right)^t ||\theta_0 - \theta_{PS}||_2$$

Now we set

$$\left(\omega\epsilon\frac{\beta}{\gamma}\right)^t ||\theta_0 - \theta_{PS}||_2 \leq \delta$$

and solve for $t$.

$$\left(\omega\epsilon\frac{\beta}{\gamma}\right)^t ||\theta_0 - \theta_{PS}||_2 \leq \delta$$

$$t\log\left(\omega\epsilon\frac{\beta}{\gamma}\right) + \log(||\theta_0 - \theta_{PS}||_2) \leq \log(\delta)$$

$$t\log\left(\omega\epsilon\frac{\beta}{\gamma}\right) \leq \log(\delta) - \log(||\theta_0 - \theta_{PS}||_2)$$

$$t\log\left(\omega\epsilon\frac{\beta}{\gamma}\right) \leq t\left(\omega\epsilon\frac{\beta}{\gamma} - 1\right) \leq \log(\delta) - \log(||\theta_0 - \theta_{PS}||_2)$$

$$t \geq (\log(\delta) - \log(||\theta_0 - \theta_{PS}||_2))\left(\omega\epsilon\frac{\beta}{\gamma} - 1\right)^{-1}$$

$$t \geq \left(1 - \omega\epsilon\frac{\beta}{\gamma}\right)^{-1}\log\left(\frac{||\theta_0 - \theta_{PS}||_2}{\delta}\right)$$

Note that this theorem is essentially just an application of the Banach fixed point theorem as $\omega\epsilon < \frac{\gamma}{\beta} \implies \omega\epsilon\frac{\beta}{\gamma} < 1$. □

While this proof provides us with conditions under which RDRO converges, it is somewhat unsatisfying as it leaves as an open question what is required for Definitions 3.5, 3.6, and 3.7 to be true. Unlike for performative prediction with risk minimization, making assumptions on our loss function does not guarantee that the distributionally robust objective will share those properties. In order to have smoothness of our objective and also robust $\epsilon$-sensitivity of the distribution map we require some sort of stability or regularity of the worst case distributions as the centre of the $f$-divergence ball moves as a function of $\theta$.

The robust $\epsilon$-sensitivity condition could possibly be equivalently expressed as a smoothness condition for the $f$-divergence ball surrounding the data generating distribution. Further exploration of these questions would likely yield further understanding of the distributionally objective, even outside of the context of performative prediction.

It could also be possible that robust $\beta$-joint smoothness is not a necessary condition for convergence of RDRO and that a convergence proof may be possible working with subgradients or directional derivatives, but if this is the case it would necessitate a different, and likely more involved, proof technique than the one used here. An alternative approach would also be to work with the dual reformulation of the DRO problem given in Theorem A.2 or to use an alternative probability distance measure rather than $f$-divergence balls that may be more amenable to this type of analysis.

These approaches come with their own complications, however, and likely require answering similar questions. We believe these are interesting questions worthy of further research, but their investigation is beyond the scope of this work. Instead, we present empirical work which demonstrates the convergence of RDRO in a variety of scenarios.

## A.5 IMPLEMENTATION

We provide a brief explanation of the implementation details for ERM and DRO. Both risk minimization and distributionally robust optimization require access to the data generating distribution, *i.e.* infinite data, which is obviously impossible, so we perform empirical risk minimization and utilize the plug-in estimator for distributionally robust optimization instead. Further, for DRO we make use of the dual formulation (Theorem A.2) and perform optimization on this objective rather than working with the primal form. Following Duchi & Namkoong (2021) and Hashimoto et al. (2018), we use $\chi^2$-divergence balls in our implementation.

In all of our experiments we use linear models trained with gradient descent. We use a fixed step-size and a fixed number of epochs for training. The logistic regression utilizes an L2-regularized cross-entropy loss function.

The dual formulation of the distributionally robust objective allows for a simple training procedure where we treat the dual variable $\eta$ as a hyperparameter. Recall that for convex losses, $\ell(\theta; Z)$, the dual formulation is jointly convex in $(\theta, \eta)$. The training procedure is thus as follows, for a given $\eta$, compute the approximate minimizer $\widehat{\theta}_\eta$

$$\underset{\theta \in \Theta}{\text{minimize}}\, \mathbb{E}(\ell(\theta; Z) - \eta)_+^2.$$

Because the dual is convex in $\eta$ we can use binary search to find the optimal dual variable $\eta^*$. The loss $(\ell(\theta; Z) - \eta)_+^2$ is merely the ReLu function applied to the usual loss with $\eta$ subtracted and then squared which allows us to train models using gradient descent methods. Hence, using binary search we train models with different values of $\eta$ until we find the optimal $\eta^*$. The optimal value, $\eta^*$, depends on the data and also the radius of our $f$-divergence ball, $\rho$. The value of $\rho$ is a hyperparameter that we must specify before training our model.

## A.6 EXPERIMENTS

### A.6.1 RRM AND RDRO FOR CREDIT DATASET

Our first experiment reproduces the experimental work from Perdomo et al. (2020). We use a dataset from a Kaggle competition titled *Give Me Some Credit*. The dataset contains relevant information for predicting credit scores. The target variable is a binary variable indicating whether or not an individual has experienced financial distress in the past two years. The original dataset contains historical information for 250,000 borrowers.

Following Perdomo et al. (2020) we balance the dataset to contain an equal number of positive and negative cases for the target variable and normalize predictor variables to have a mean of zero and variance of one. The reduced dataset contains 18,358 entries.

We train L2-regularized logistic regression models on the data for both ERM and DRO. Both models are trained with stochastic gradient descent with a fixed step-size of $\alpha = 0.03$ for 5000 epochs. The step-size and number of epochs were chosen empirically to approximately replicate performance from Perdomo et al. (2020) on the base distribution. Additionally, a fixed value of $\rho$ was chosen as a radius of the $\chi^2$-divergence ball for DRO. The value of $\rho$ was chosen so that the accuracy of the DRO model on the full dataset was significantly, but not drastically, different than that of the ERM model.

In order to add a performative element to the prediction problem, we identify 3 of the 10 features as strategic features which will be altered as a function of the parameters of our model. Strategic classification is a particular instance of performative prediction in which individuals adversarially adjust their features to maximize the likelihood of classification to the positive class. Strategic classification has been explored in relation to fairness concerns in previous work Hardt et al. (2016a); Bruckner et al. (2012); Miller et al. (2020); Milli et al. (2019); D'Amour et al. (2020); Hu et al. (2019).

As described in Perdomo et al. (2020) and Hardt et al. (2016a), we assume individuals have linear utilities $u(\theta, x) = -\langle \theta, x \rangle$ and quadratic costs $c(x', x) = \frac{1}{2\epsilon}||x' - x||_2^2$. The constant $\epsilon$ controls the cost individuals incur by altering their features. Individuals thus pay a cost to manipulate their features in order to minimize the likelihood of the model predicting that they will default on their loan, but are unable to change the true outcome, $y \in \{0, 1\}$, of whether or not they default. Given linear utilities and quadratic costs as described here, the individuals' best response is to manipulate their features as

$$x'_S = x_S - \epsilon\theta_S,$$

where $x_S, x'_S\, \mathbb{R}^{|S|}$ and $|S|$ is the number of strategic features. The explanation of why this distribution map is $\epsilon$-sensitive can be found in Perdomo et al. (2020).

The procedure for updating the data according to the distribution map for strategic classification is explained in the box below.

---

**Input:** Base distribution $P$, a classifier $f_\theta$, a cost function $c$, a utility function $u$. **Sampling procedure for $\mathcal{D}(\theta)$:**

1. Sample $(x, y) \sim P$
2. Compute best response $x_{BR} \leftarrow \arg\max_{x'} u(x', \theta) - c(x', x)$
3. Output sample $(x_{BR}, y)$

---

Using a logistic regression classifier and the strategic classification distribution map sampling procedure outlined above, we run ERM and DRO for 30 iterations on our dataset with values of $\epsilon \in \{0.1, 1, 10, 100\}$. We observe similar convergence behaviour for both ERM and DRO. As the value of $\epsilon$ grows, the inequality $\epsilon < \frac{\gamma}{\beta}$ no longer holds, meaning that the conditions of Theorems A.9 and 3.8 are not satisfied and we do not necessarily have a contraction mapping. We plot the normalized distance between values of $\theta$ for successive iterations of ERM and DRO in Figures 1 and 2. The distance between iterates is calculated as

$$\frac{1}{||\theta_S||_2} \cdot ||\theta_{t+1} - \theta_t||_2,$$

where $\theta_S$ is the value of $\theta_0$ on the strategic features.

### A.6.2 BUILDING INTUITION THROUGH SIMPLE EXAMPLES

We now investigate the convergence of ERM and DRO on simple synthetic datasets to build intuition for the behaviour of the two algorithms. We move away from the assumptions required for our general theoretical guarantees for convergence to a fixed point and experiment with distribution maps that are not necessarily $\epsilon$-sensitive. We begin with a regression problem and then investigate a classification problem.

**Regression**

We start with a simple mean prediction task with data sampled from a mixture of two univariate Gaussians, $X_A \sim \mathcal{N}(\mu_A, \sigma^2)$ and $X_B \sim \mathcal{N}(\mu_B, \sigma^2)$. 80% of the data is sampled from $X_A$ and 20% from $X_B$, *i.e.* $X = 0.2X_A + 0.8X_B$. We train a linear regression model using gradient descent with backtracking line search to predict the mean of the data. We initialize our data with values $\mu_A = 4$, $\mu_B = 4$, and $\sigma^2 = 0.01$. We choose a small value of $\sigma^2$ so that the variance and finite samples have only a small impact on the convergence behaviour of ERM and DRO. We select an $f$-divergence ball radius of $\rho = 4$ for DRO.

We use $\theta$ to denote the learned parameter of the model, $\mu$ to denote the true mean of the data generating distribution, and $\mu_A$ and $\mu_B$ to denote the true means of the $X_A$ and $X_B$. In other words, $\theta = \hat{\mu}$ is the estimated mean from the model. Where necessary, we indicate with subscripts ERM and DRO to indicate which model we are referring to.

We investigate three different distribution maps which we call $\mathcal{D}_0$, $\mathcal{D}_1$, and $\mathcal{D}_2$. For each map the means of the normal distributions from which we sample are adjusted as a function of $\theta$. Hence, the induced distribution is

$$\mathcal{D}_i(\theta) = 0.2\mathcal{N}(\mathcal{D}_i^A(\theta), \sigma^2) + 0.8\mathcal{N}(\mathcal{D}_i^B(\theta), \sigma^2).$$

These distribution maps were chosen because they are simple to understand, but reveal important differences in the way ERM and DRO behave when their predictions alter the distribution on which they are learning. The distribution maps are specified in Table 3.

| $\mathcal{D}_0(\cdot)$ | $D_1(\cdot)$ | $\mathcal{D}_2(\cdot)$ |
|---|---|---|
| $\mathcal{D}_0^A(\theta) = \mathcal{N}(\theta, \sigma^2)$ | $\mathcal{D}_1^A(\theta) = \mathcal{N}(\mu_A, \sigma^2)$ | $\mathcal{D}_2^A(\theta) = \mathcal{N}(2\theta, \sigma^2)$ |
| $\mathcal{D}_0^B(\theta) = \mathcal{N}(\frac{\theta}{2}, \sigma^2)$ | $\mathcal{D}_1^B(\theta) = \mathcal{N}(\frac{\theta}{2}, \sigma^2)$ | $\mathcal{D}_2^B(\theta) = \mathcal{N}(\frac{\theta}{2}, \sigma^2)$ |

Table 3: Distribution maps for mean-prediction experiment.

The distribution map determines the evolution of the distribution over the data at each iteration of deploying and learning our model, which in turn changes the learned $\theta_t = \hat{\mu}_t$. Despite starting from

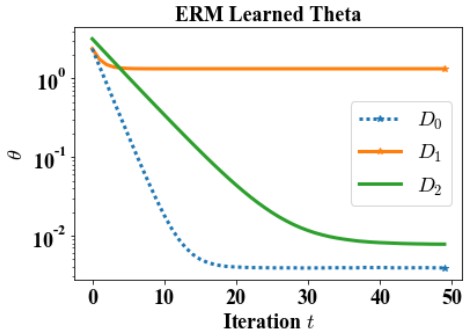
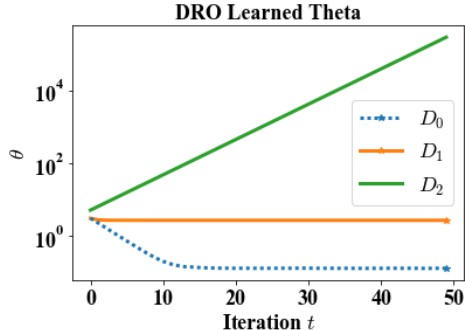

Figure 8: Learned values of $\theta$ for ERM.

Figure 9: Learned values of $\theta$ for DRO.

the same initial distribution in each case, the induced distributions vary widely. This demonstrates the importance of performativity and illustrates why a static supervised learning framework fails to adequately capture the complex dynamics of prediction problems which involve performativity.

We run the experiment for 50 iterations, where one iteration involves training a model on data sampled from the current distribution and then updating the distribution via the distribution map. We summarize the evolution of the learned parameter $\theta_t$ over time for both ERM and DRO in Figures 8 and 9. Given the simplicity of the learning problem, the learned values of $\theta$ closely approximate true mean of the distribution, $\mu$, over time. The approximate values to which ERM and DRO converge are given in Table 4.

It is interesting to note that even with this simple mean prediction task we observe significant differences in the behaviour of ERM and DRO over time. For all three distribution maps we observe ERM's tendency to focus on the majority group over the minority group. Recall that 20% of the data is sampled from $X_A$ and 80% from $X_B$ so we refer to group A as the minority group and group B as the majority group. For $\mathcal{D}_0$, ERM quickly converges toward zero as group B's mean evolves as $\mu_B = \frac{\theta}{2}$. Interestingly, however, it does not converge to zero, but instead remains fixed at approximately $\theta = 0.004$ even though $\theta = 0$ is the performatively optimal value.

DRO displays similar behaviour for this distribution map, but does not converge as close to zero. As noted, performative stability and performative optimality are distinct solution concepts, and performatively optimal points only coincide with performatively stable points in specific settings.

| Method | $\mathcal{D}_0(\cdot)$ | $\mathcal{D}_1(\cdot)$ | $\mathcal{D}_2(\cdot)$ |
|---|---|---|---|
| ERM | $\theta_{ERM} = 0.004$ | $\theta_{ERM} = 1.33$ | $\theta_{ERM} = 0.008$ |
| DRO | $\theta_{DRO} = 0.128$ | $\theta_{DRO} = 2.66$ | $\theta_{DRO} = \infty$ |

Table 4: Values to which $\theta$ converges for ERM and DRO.

The second distribution map, $\mathcal{D}_1$, reveals differences in ERM and DRO that are relevant for fairness. For $\mathcal{D}_1$, the mean of the majority group again evolves as $\mu_B = \frac{\theta}{2}$, but this time the minority group's mean remains unchanged with $\mu_A = 4$. As ERM averages the loss over all data points, it converges to a predicted mean much closer to $\mu_B$ than to $\mu_A$. DRO, on the other hand, converges to a value that balances performance on prediction of the means of both minority and majority groups. In fact the distance from the true means of group A and group B is almost identical.

$$|\theta_{DRO} - \mu_A| = |2.66 - 4| = 1.34$$
$$|\theta_{DRO} - \mu_B| = |2.66 - 1.33| = 1.33$$

ERM, however, is a much better predictor of the global mean of the data. This simple example illustrates a trade-off we can expect between ERM and DRO in terms of fairness versus global

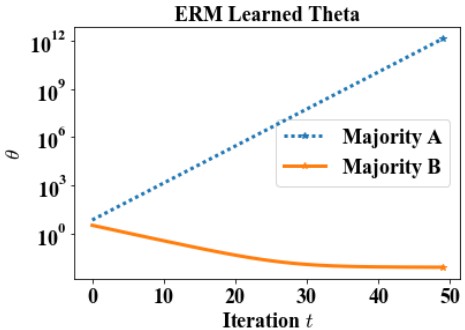 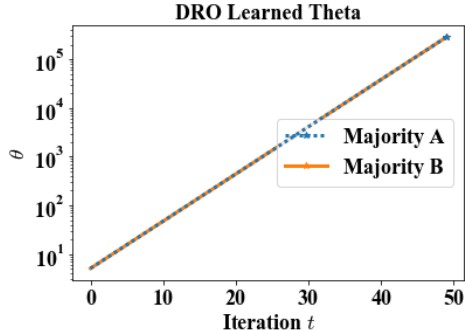

Figure 10: Learned values of $\theta$ for ERM with $\mathcal{D}_2$.

Figure 11: Learned values of $\theta$ for DRO with $\mathcal{D}_2$.

performance.

$$\mu_{ERM} = 0.2\mu_A + 0.8\mu_B = (0.2)(4) + (0.8)(0.665) = 1.332$$
$$\mu_{DRO} = 0.2\mu_A + 0.8\mu_B = (0.2)(4) + (0.8)(0.1.33) = 1.864$$

The final distribution map, $\mathcal{D}_2$, demonstrates that ERM and DRO can behave entirely differently under some circumstances. For this distribution map ERM converges to a similar value to that under $\mathcal{D}_0$, that is $\theta_{ERM} = 0.008$. This makes some sense intuitively as $\mu_A = 2\theta$ whereas for $\mathcal{D}_0$ we had $\mu_A = \theta$. DRO, on the other hand, diverges with $\theta_{DRO}$ going to infinity. The reason for this is that for the first iteration DRO learns a value of $\theta_{DRO}$ that is larger than 4. Similarly, for each successive iteration the learned value of $\theta_{DRO}$ gets larger which in turn causes the true means of the data to grow as a function of the learned $\theta_{DRO}$. Interestingly, if you swap the majority and minority groups the behaviour of DRO remains nearly identical, while ERM diverges at a faster rate than DRO. This again demonstrates how DRO treats minority and majority groups similarly, while ERM learns a function that prioritizes performance on majority groups. We illustrate this in Figures 10 and 11

**Classification**

We now move to a classification task and analyze the behaviour of logistic regression classifiers trained using ERM and DRO. The classification task is more complex than the simple mean prediction task, so for this experiment we analyze only the static supervised learning setting in order to reduce complexity and elucidate the relevant differences between ERM and DRO.

Our data is generated from bivariate Gaussian distributions and the label, $y$, of a given data point is 1 if the sum of its features are greater than the sum of the means of the two Gaussian distributions from which we draw samples and 0 otherwise. As with the regression experiments, we have two subgroups within our data, A and B. We vary the proportion of samples from each subgroup in the experiments. Precisely, the data generating process is as follows:

$$X_A \sim \mathcal{N}(\boldsymbol{\mu}_A, \boldsymbol{\Sigma}_A)$$
$$X_B \sim \mathcal{N}(\boldsymbol{\mu}_B, \boldsymbol{\Sigma}_B)$$
$$X = c_A X_A + c_B X_B \quad c_A, c_B \in (0,1), \text{ and } c_A + c_B = 1$$

where

$$\boldsymbol{\mu}_A = \begin{bmatrix} \mu_A^1 \\ \mu_A^2 \end{bmatrix}, \qquad \boldsymbol{\mu}_B = \begin{bmatrix} \mu_B^1 \\ \mu_B^2 \end{bmatrix}, \qquad \boldsymbol{\Sigma}_A = \begin{bmatrix} \sigma_A^1 & 0 \\ 0 & \sigma_A^2 \end{bmatrix}, \qquad \boldsymbol{\Sigma}_B = \begin{bmatrix} \sigma_B^1 & 0 \\ 0 & \sigma_B^2 \end{bmatrix}.$$

And for a data point, $x = [x_i^1, x_i^2]^T$, with $i \in \{A, B\}$,

$$y = \begin{cases} 0 & \text{if } x_i^1 + x_i^2 \leq \mu_i^1 + \mu_i^2 \\ 1 & \text{if } x_i^1 + x_i^2 > \mu_i^1 + \mu_i^2 \end{cases}.$$

Hence, if $\boldsymbol{\mu}_A \neq \boldsymbol{\mu}_B$, the data is not linearly separable and the logistic regression model must trade off performance across the two subgroups. We provide scatter plots of samples from the data generating

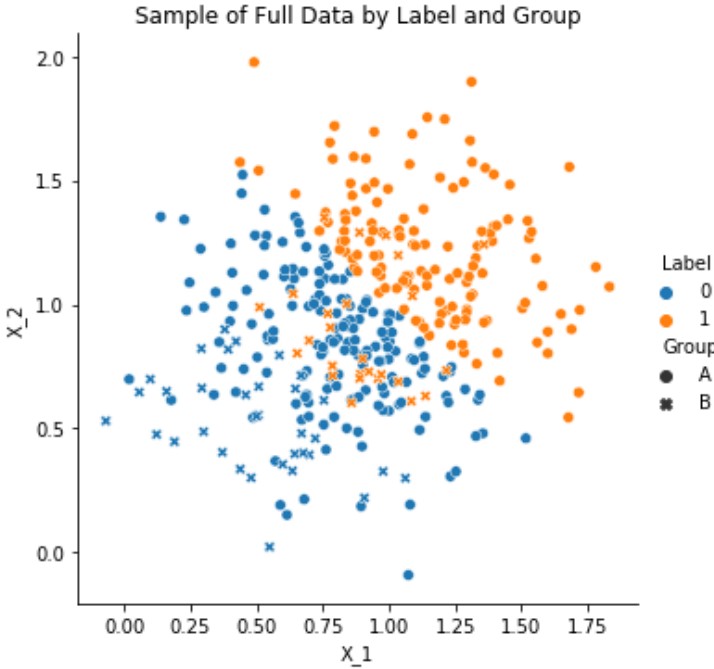

Figure 12: Sample of 360 data points from the data generating distribution with $c_A = 0.8$ and $c_B = 0.2$.

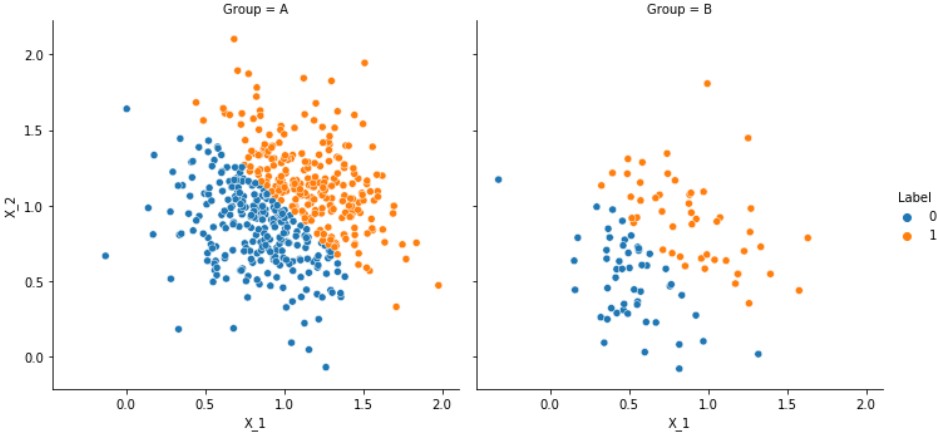

Figure 13: Sample of 600 data points from the data generating distribution with $c_A = 0.8$ and $c_B = 0.2$.

distribution below with $\mu_A^i = 1$ and $\mu_B^i = 0.7$, $\sigma_A^i = \sigma_B^i = 0.1$ for $i \in \{0, 1\}$, and $c_A = 0.8$, $c_B = 0.2$. Figure 12 contains a sample of 360 data points coloured by the value of their target variable, with crosses representing data points belonging to group B and circles representing data points belonging to group A. Figure 13 contains a sample of 500 and 100 data points respectively from group A and group B coloured by the value of their target variable.

In Figure 12 we can see that the data is not linearly separable, as some members of group B who belong to the positive class have features that place them lower than the threshold for positive classification for group A. Although this dataset is extremely simple, it is characterized by a feature that represents a central concern for fairness in machine learning, namely that the conditional probability distributions, $P(y|x)$, are significantly different for distinct subsets of the data. This example

is intended to represent a simplified abstract instance of a population with majority and minority subgroups in order to see how the behaviour of ERM and DRO differ in this circumstance.

We generate three distinct datasets on which to train our algorithms, each made up of differing proportions of the two subgroups A and B. Each dataset contains a sample of 10,000 data points, with samples distributed according the values of $c_A$ and $c_B$. The accuracy of the models on the three datasets is summarized in Tables 5 and 6 below. For both ERM and DRO we use L2-regularized logistic regression trained with stochastic gradient descent. The step-size for all algorithms is fixed at 0.05 and we train for 15,000 epochs.

Models trained with the distributionally robust objective have the additional complication that we must specify a value for the radius of the $\chi^2$-divergence ball, *i.e.* $\rho$. The larger the value of $\rho$, the more we can expect a DRO model to differ from an ERM model because as the $\chi^2$-divergence ball grows, the worst case distribution can be further and further from the data generating distribution. Conversely, in the limit as $\rho \to 0$, we recover ERM as the $\chi^2$-divergence ball shrinks to contain only the data generating distribution.

Choosing the value of $\rho$ is a challenging decision, as the performance of a model varies significantly as $\rho$ changes. If one has access to demographic information, it is possible to conduct a grid search over possible $\rho$ values in order to find a value that results in a model with the desirable fairness properties. Doing this, however, largely defeats the purpose of DRO. As explained earlier, a central advantage to using DRO rather than some fairness constrained optimization technique is that DRO does not require access to demographic information. In this experiment we work directly with the dual formulation of DRO and set $\eta = 0.56$. This value was chosen empirically to achieve relatively uniform accuracy across group A and group B for an 80/20 split between the two subgroups. As the values of $c_A$ and $c_B$ change, we can see that the performance of DRO changes for a given value of $\eta$ and hence $\rho$, as $\eta^*$ depends on $\rho$.

| Group | $[c_A = 0.6, c_B = 0.4]$ | $[c_A = 0.8, c_B = 0.2]$ | $[c_A = 0.95, c_B = 0.05]$ |
|---|---|---|---|
| A | 0.797 | 0.907 | 0.966 |
| B | 0.701 | 0.652 | 0.592 |
| All Data | 0.759 | 0.856 | 0.948 |

Table 5: Accuracy by Group for ERM.

| Group | $[c_A = 0.6, c_B = 0.4]$ | $[c_A = 0.8, c_B = 0.2]$ | $[c_A = 0.95, c_B = 0.05]$ |
|---|---|---|---|
| A | 0.665 | 0.751 | 0.780 |
| B | 0.869 | 0.744 | 0.766 |
| All Data | 0.747 | 0.750 | 0.780 |

Table 6: Accuracy by Group for DRO.

We first examine the performance of ERM (Table 5). As the data is not linearly separable, ERM must learn a decision boundary that trades off performance between the two subgroups. Because the ERM objective treats the loss on each data point equally, the model learns a decision boundary that is more accurate for the majority group than for the minority group. This discrepancy in accuracy of predictions worsens the smaller the majority group is. For instance, when 95% of the data comes from group A, the logistic regression model trained with an ERM objective achieves 96% accuracy on group A members, but only 59.2% accuracy on group B members.

Models trained with a DRO objective behave much differently. For the 80/20 split and 95/5 split, the DRO models learn relatively fair decision boundaries, effectively balancing performance on both subgroups A and B. For the 60/40 split, however, DRO actually learns a model that performs significantly better on the minority group than the majority group. This model is in a sense discriminatory against the majority group. This is the result of the radius of the $\chi^2$-divergence ball being too large and the model thus overly focusing on a worst case distribution.

As mentioned above, the correct choice of $\rho$ is not necessarily obvious, but it greatly impacts the performance of the model. Duchi *et al.* provide some recommendations and heuristics for choosing

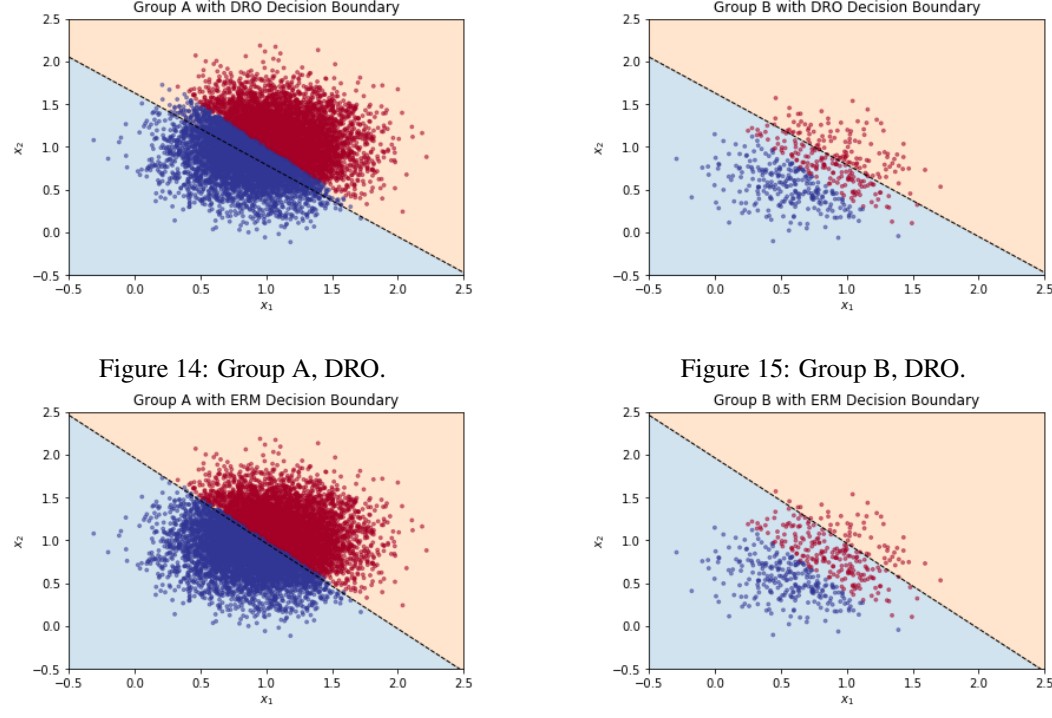

Figure 14: Group A, DRO.

Figure 15: Group B, DRO.

Figure 16: Group A, ERM.

Figure 17: Group B, ERM.

Figure 18: Decision boundaries for ERM and DRO classifiers with samples from groups A and B with $c_A = 0.95$ and $c_B = 0.05$. Shading indicates predicted label and data point colour indicates true label.

values of $\rho$ in Duchi & Namkoong (2021). Along with altering the value of $\rho$, one can change the $f$-divergence ball by varying the the value of $k$ for $f$-divergences in the Cressie-Read family. This has a similar effect to changing the value of $\rho$.

In Figures 14, 15, 16, and 17 we plot data from groups A and B for data generated with $c_A = 0.95$, $c_B = 0.05$ with the learned decision boundary from a DRO and ERM model, respectively, overlaid. The background shading indicates the predicted label with blue representing $\hat{y} = 0$ and red representing $\hat{y} = 1$, while the colour of the data points indicate their true label.

Neither the regression nor the classification experiments in this section are designed to be realistic representations of real-world applications, but rather are intended to provide a simple setting in which to investigate important differences in the behaviour of models trained with ERM versus DRO objectives. In the next section we investigate a slightly more complex classification task in the performative prediction setting and examine how DRO and ERM may impact fairness considerations when deploying a model influences the distribution on which it is making predictions.

### A.6.3 FAIRNESS AND RDRO

The data generating process is as follows:

$$X_A \sim \mathcal{N}(\boldsymbol{\mu}_A, \boldsymbol{\Sigma}_A)$$
$$X_B \sim \mathcal{N}(\boldsymbol{\mu}_B, \boldsymbol{\Sigma}_B)$$
$$X = c_A X_A + c_B X_B \quad c_A, c_B \in (0, 1), \text{ and } c_A + c_B = 1$$

where

$$\boldsymbol{\mu}_A = \begin{bmatrix} \mu_A^1 \\ \vdots \\ \mu_A^{10} \end{bmatrix}, \quad \boldsymbol{\mu}_B = \begin{bmatrix} \mu_B^1 \\ \vdots \\ \mu_B^{10} \end{bmatrix}, \quad \boldsymbol{\Sigma}_A = \begin{bmatrix} \sigma_A^1 & \cdots & 0 \\ \vdots & \ddots & \vdots \\ 0 & \cdots & \sigma_A^{10} \end{bmatrix}, \quad \boldsymbol{\Sigma}_B = \begin{bmatrix} \sigma_B^1 & \cdots & 0 \\ \vdots & \ddots & \vdots \\ 0 & \cdots & \sigma_B^{10} \end{bmatrix}.$$

And for a data point, $x = [x_i^1, \ldots, x_i^{10}]^T$, with $i \in \{A, B\}$,

$$y = \begin{cases} 0 & \text{if } x_i^1 + \cdots + x_i^{10} \leq \mu_i^1 + \cdots + \mu_i^{10} \\ 1 & \text{if } x_i^1 + \cdots + x_i^{10} > \mu_i^1 + \cdots + \mu_i^{10} \end{cases}.$$

For our experiment we set the parameters of the data generating process as

$$\mu_A^i = 1 \qquad\qquad \sigma_A^i = 0.1 \qquad\qquad c_A = 0.8$$
$$\mu_B^i = 0.8, \qquad\qquad \sigma_B^i = 0.1, \qquad\qquad c_B = 0.2.$$

These parameters were selected as an attempt to capture the notion that an underprivileged minority group may have features that make them appear to be less qualified, despite having the same proportion of qualified individuals as a dominant majority group. The data generating process obviously represents this at a high level of abstraction and is much less complex than most real-world applications. With that said, we believe the data generating process effectively encapsulates this abstracted characteristic that is central to concerns for learning fair models. Examples of the type of situation that this experiment is intended to represent are college admissions or hiring, where an underprivileged minority group may not, on average, have CVs that appear as impressive as their peers from the majority group due to a lack of opportunity, but are nevertheless equally qualified for the school or job.

We once again examine a strategic classification scenario, as in experiment 1, but our data now contains subgroups, allowing us to analyze the impact of ERM and DRO on fairness and model performance in the performative setting. We examine four different distribution maps by varying the parameter $\epsilon \in \{0.01, 0.25, 0.5, 10\}$ and set 5 of the 10 features to be strategic (*i.e.* manipulable).

For both ERM and DRO we train L2-regularized logistic regression models with $\lambda = 0.0001$. We use stochastic gradient descent with a fixed step-size of $\alpha = 0.2$ and train for 8000 epochs on samples of 1,200 data points. We use a fixed radius $\rho$ of the $\chi^2$-divergence ball for DRO. All parameters were chosen empirically to give good performance on the base distribution.

First, we examine the convergence behaviour of both algorithms by plotting the normalized distance between successive iterates of the learned parameters, $\theta_t$, in Figures 19 and 20. We observe that ERM does not converge for any value of $\epsilon$, while DRO converges for only $\epsilon = 0.01$. It is unclear why the algorithms do not converge for other values of $\epsilon$. The failure to converge could be related to using a fixed, rather than decaying step-size, or because the conditions for the contraction mapping are not met (recall that the conditions are sufficient, but not necessary for convergence of a particular instance). For the three smaller values of $\epsilon$, both ERM and DRO converge to a small neighbourhood, whereas for $\epsilon = 10$, neither ERM nor DRO exhibit any convergence.

To demonstrate the effect of the convergence, or lack thereof, of $\theta$ on the model's performance, we plot the average supervised and performative L2-regularized binary cross-entropy loss for ERM and DRO for $\epsilon = 0.5$ and $\epsilon = 10$ in Figure 21. The blue lines indicate the optimization phase and the green lines indicate the effect of the distribution shift after the classifier deployment. That is, the dots at the end of a green dotted line represent performative loss, while the dots at the end of a blue line represent supervised loss. In the plots we can see that even though ERM and DRO do not converge for $\epsilon = 0.5$, the models quickly achieve relatively stable loss on the classification task. Note that the average L2-regularized binary cross-entropy loss is the objective for which ERM is optimizing, but not the objective for which DRO is optimizing.

Given that this experiment is a balanced binary classification task, the metric in which we are principally interested is accuracy. Furthermore, as we are interested in comparing the fairness properties of DRO and ERM, it is important to analyze the performance of the models on the subgroups within the population, as well as the population as a whole. In Figures 22 and 23 we plot the the accuracy of the two models for $\epsilon = 0.5$. We see that the performative accuracy of ERM initially degrades significantly, before quickly converging to approximately 84% on the full population.

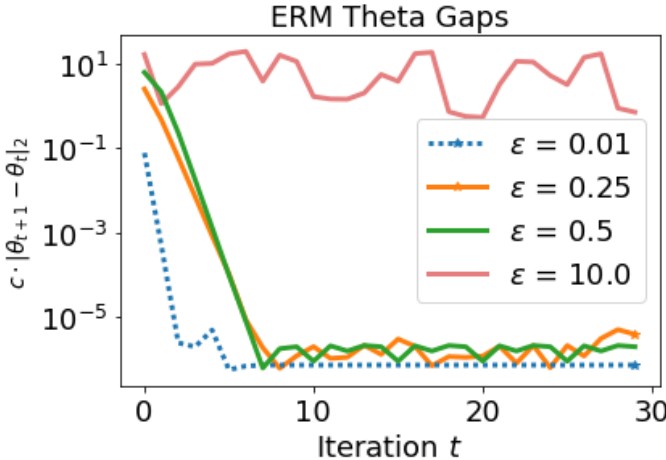

Figure 19: Plot of the normalized distance between successive values of $\theta$ for ERM.

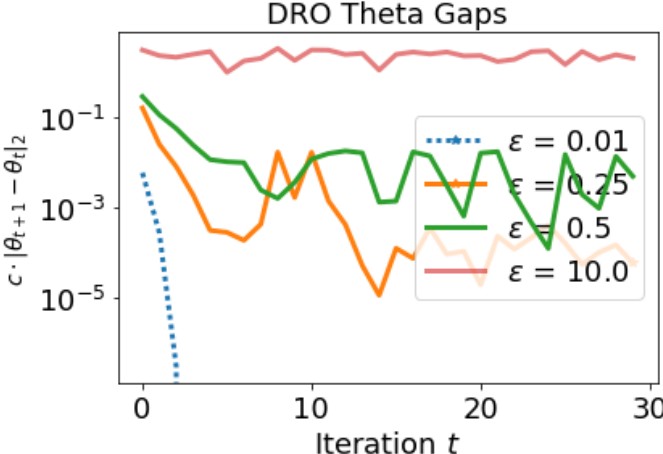

Figure 20: Plot of the normalized distance between successive values of $\theta$ for DRO.

Conversely, the performative accuracy for DRO is actually initially higher than the accuracy on the distribution on which the model was trained, but it also converges relatively quickly to an accuracy of approximately 72.5% on the full population. This is once again due to DRO having a different optimization objective than ERM. The average cross-entropy loss acts as a surrogate loss for the 0-1 loss and hence aims to maximize accuracy on the full dataset. The objective used for DRO focuses on the tails of the distribution and therefore does not maximize accuracy across the full dataset. The improvement of the DRO model's accuracy after the distribution shift is not an inherent property of DRO, but rather is specific to this particular dataset and distribution map. We see in Figure 22 at iteration $t = 2$ that ERM also occasionally achieves better performance after distribution shift. In general, the improvement or degradation of accuracy depends on the learned parameters of the model, the distribution map, the data, and the loss function.

In Figure 25 we plot the accuracy of ERM and DRO for the subgroups A and B for $\epsilon = 0.5$. We can see from these plots that the fairness properties of ERM and DRO are preserved in the performative prediction setting.

ERM converges to an accuracy of approximately 90% on group A and only approximately 54% on group B. DRO, on the other hand, converges to much more equal accuracy across the two subgroups, achieving an accuracy of approximately 74% for group A and approximately 66% for group B. We summarize the accuracies to which ERM and DRO converge in Tables 7 and 8. We do not include

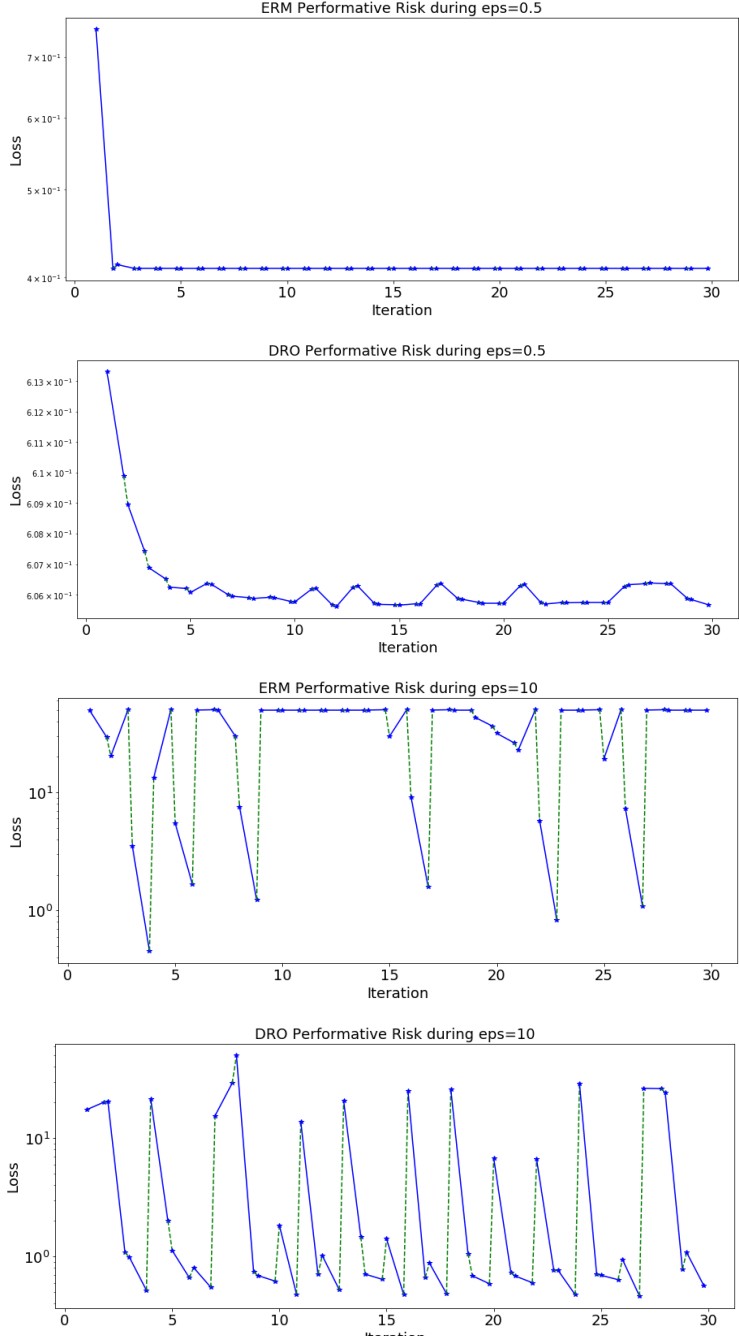

Figure 21: Plots of the average L2-regularized binary cross-entropy supervised and performative loss.

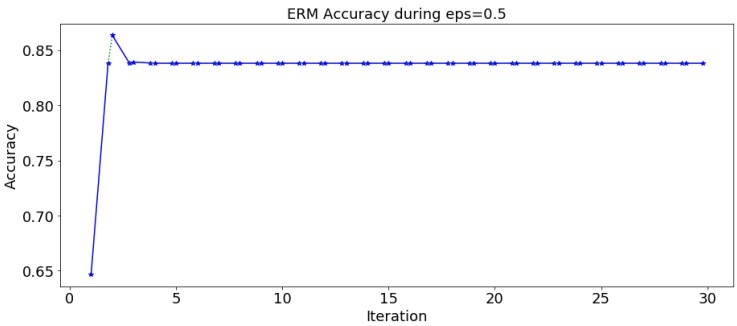

Figure 22: ERM

Figure 23: DRO

Figure 24: ERM and DRO accuracy on the full population across successive iterations.

the accuracy for $\epsilon = 10$ because neither ERM nor DRO converged to a small enough neighbourhood, and therefore did not converge in accuracy. We see that the algorithms converge to similar accuracy values for all the values of $\epsilon$.

This result, while perhaps not that surprising, is important, as it demonstrates that not only does DRO exhibit similar convergence behaviour to ERM, but DRO converges to fair fixed points, whereas ERM converges to discriminatory fixed points in the presence of heterogeneous data composed of minority and majority subgroups. Recall also that DRO is not given access to group information, but still learns to achieve more uniform performance across subgroups, as it is attempting to minimize the worst case loss across all probability distributions within the $\chi^2$-divergence ball surrounding the data generating distribution.

ERM Performative Accuracy

| Group | $\epsilon = 0.01$ | $\epsilon = 0.25$ | $\epsilon = 0.5$ |
|---|---|---|---|
| A | 0.893 | 0.896 | 0.898 |
| B | 0.540 | 0.540 | 0.540 |
| All Data | 0.834 | 0.837 | 0.838 |

Table 7: Accuracy by Group for ERM after 30 iterations.

## A.7 FINAL THOUGHTS

We began this work with a discussion of why research into fairness in machine learning is such an urgent issue and we would like to reemphasize this point. Machine learning is no longer a niche research area. It has become a fundamental driving force of the global economy and now attracts billions of dollars in investment around the world Abdallat (2021); Balakrishnan et al. (2020). This

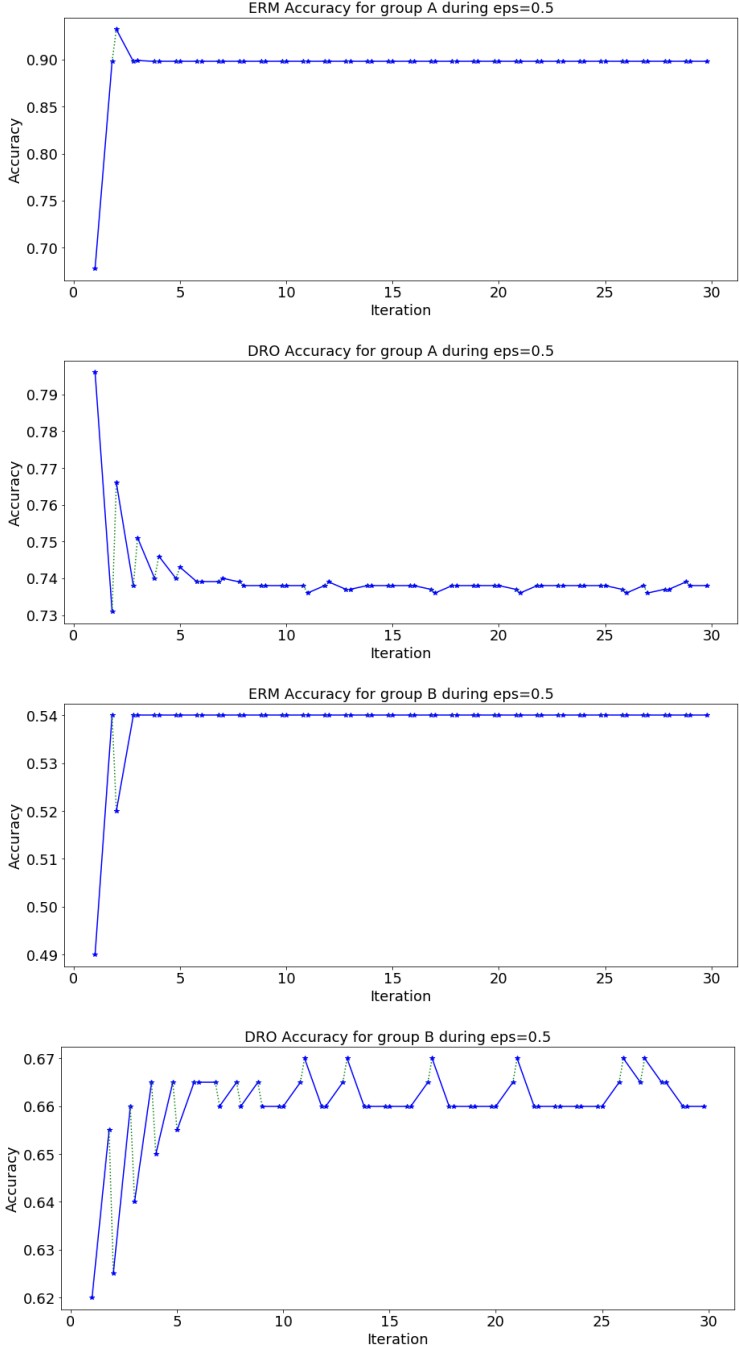

Figure 25: ERM and DRO accuracy on subgroups A and B across successive iterations.

DRO Performative Accuracy

| Group | $\epsilon = 0.01$ | $\epsilon = 0.25$ | $\epsilon = 0.5$ |
|---|---|---|---|
| A | 0.687 | 0.710 | 0.738 |
| B | 0.670 | 0.660 | 0.660 |
| All Data | 0.684 | 0.701 | 0.725 |

Table 8: Accuracy by Group for DRO after 30 iterations.

change has resulted in enormous opportunity for individuals with expertise in machine learning and it appears that this trend will only increase in the coming years.

The reality, however, is that this emergence of AI as an economic powerhouse has not occurred in a particularly responsible manner. Machine learning researchers and engineers have benefited greatly from the investment in AI and the community has a responsibility to ensure that the future development of the field aligns with and supports universal rights, freedoms, and values. Chief among these responsibilities is ensuring that the development and deployment of machine learning models do not harm the most vulnerable among us. We hope that this work contributes to this endeavor.

