# OpenReview forum: "Long Term Fairness via Performative Distributionally Robust Optimization"
_ICLR.cc/2023/Conference — Submitted to ICLR 2023_

### Official Review · Reviewer_oy4Y · 2022-10-24

**Confidence:** 3
**Correctness:** 3
**Technical Novelty And Significance:** 2
**Empirical Novelty And Significance:** 2
**Recommendation:** 5

**Clarity, Quality, Novelty And Reproducibility:**

**Quality** The paper is technically sound. However, some key information is missing when presenting the main results (Def. 3.5 and Def. 3.6).

**Clarity** The paper is well written. But the experimental setup is hard to follow.

**Novelty** The paper is a combination of well-known techniques, and it does not clearly state the novel of technical parts.

**Reproducibility** The paper does not provide any supplementary material for reproducibility in the experiments.

**Strength And Weaknesses:**

Strengths:
1. The paper is well-written and well-motivated. In particular, the paper clearly justifies the proposed method by listing the limitations of the prior methods.
2. The high-level background knowledge of DRO and performative prediction is clear and easy to follow.
3. The empirical results demonstrate the proposed method is effective in learning fair ML models.

Weaknesses:
1. The paper does not clearly state the technical novelty of the proposed method and misses some essential information when presenting the main results. For example, the authors state, "The supremum over the uncertainty set introduces additional complications necessitating novel definitions and an *altered* proof for convergence to a performatively stable model." However, there are no details about the technical challenges until we dive into the proofs. I suggest the author provide some information in the main text as well. Besides, Definitions 3.5 and 3.6 seem incomplete, making it difficult to follow the proofs.
2. Experiments are hard to read, especially for the introduction of strategic classification. I know little about strategic classification. Could you please provide more background information (e.g., feature-changing dynamics)? Besides, I am also interested to see some visualization of how the features change over time, given DRO and ERM.
3. Missing references to reinforcement learning and online learning for fairness. The authors mention bandits, reinforcement learning, and online learning in "Conclusions ."There is a line of work focused on long-term fairness in this area, to name a few [1-5].

[1] Fairness in Reinforcement Learning

[2] Algorithms for Fairness in Sequential Decision Making

[3] Towards Return Parity in Markov Decision Processes

[4] On preserving non-discrimination when combining expert advice

[5]  Equal opportunity in online classification with partial feedback

**Summary Of The Paper:**

The paper presents repeated distributionally robust optimization (RDRO), a theoretical framework that extends the performative prediction with distributionally robust optimization. The authors provide a convergence analysis of RDRO and empirically demonstrate its implications for fair ML.

**Summary Of The Review:**

The paper is technically sound and well-written. However, the paper's novelty is unclear, and the presentation still needs improvement.

---

### Official Review · Reviewer_MxRz · 2022-10-25

**Confidence:** 3
**Correctness:** 2
**Technical Novelty And Significance:** 2
**Empirical Novelty And Significance:** 2
**Recommendation:** 3

**Clarity, Quality, Novelty And Reproducibility:**

Overall, I think the paper can benefit a lot from a clearer presentation of the central claim and the supporting arguments. In terms of the technical contribution, I am not sure how the presented results can illustrate the advantage of RDRO for the purpose of improving long-term fairness.

**Strength And Weaknesses:**

## Strength

The paper makes an effort to reflect on previous fairness notions by listing four limitations of some previously proposed fairness notions. The paper also attempts to solve some of the aforementioned issues by considering performative prediction with distributionally robust objectives.

## Weakness

### 1. w.r.t. the listed four limitations

I agree with authors that we need to reflect on previously proposed fairness notions. However, I do not agree with the way limitations are presented. For Limitation 1 (as they also acknowledged), there is no one-size-fits-all solution, therefore, we should not expect the equivalence between fairness notions in the first place, i.e., Limitation 1 is not problematic. For Limitation 2, fairness has actually been studied in dynamic settings, and fairness notions can be applied in the dynamic setting (e.g., the references in the survey paper by Zhang and Liu, 2020 "Fairness in Learning-Based Sequential Decision Algorithms: A Survey"). For Limitation 3, while an accurate estimation of fairness violation prefers the availability of demographic information, we can in certain cases make use of a subset (instead of the whole set) with ground truth to get accurate unfairness estimation. For Limitation 4, intersectionality is not ignored in more recent fairness notions, e.g., Fairness Gerrymandering characterized by Kearns et al., (2018), subgroup fairness definitions.

### 2. the connection of presented RDRO results and the listed limitations

While I understand the fact that the paper considers repeated distributionally robust optimization (RDRO) and presents convergence analysis, I am not sure how these results connect to the listed limitations of some previous fairness notions. In particular, how to parse those results in the context of long-term fairness, as claimed in the paper?

**Summary Of The Paper:**

The paper lists four limitations of current fairness studies and proposes to address (some of) them by considering performative prediction with distributionally robust objectives.

**Summary Of The Review:**

The paper lists four limitations of current fairness studies (which I hesitate to agree with in the current form), and proposes to address some of them via RDRO (which I am not sure how to parse its connection to previously listed limitations). The paper can benefit from content organizations, so that readers can understand how their presented results support their central claim.

====Post Rebuttal====

I acknowledge that I have read reviewers' comment, authors' responses, and have incorporated them in evaluation.

---

### Official Review · Reviewer_5ptm · 2022-10-25

**Confidence:** 4
**Correctness:** 4
**Technical Novelty And Significance:** 3
**Empirical Novelty And Significance:** 4
**Recommendation:** 8

**Clarity, Quality, Novelty And Reproducibility:**

The method is novel and sound. The presentation of the paper is clear, and it is easy to read.
There is a clarity issue, though. The authors extensively use "<<" symbol throughout the paper without first defining it.

**Strength And Weaknesses:**

Strengths:
- Novel idea on long-term fairness
- Very interesting analysis of the behavior of performative prediction under distributionally robust optimization.
- The benefits of the method are demonstrated for long-term fairness.

Weakness:
- I would suggest the author also compare the method with the standard group fairness model (any of them, preferably an in-processing model), to show that directly using this 'classical' fairness model will not perform well in the long term due to distribution shift, etc.
- Can the author discuss the possibility that the fair distribution is not in the uncertainty set? How should \rho be set such that the desired fair distribution is in the DRo's uncertainty set?

**Summary Of The Paper:**

The authors address the problem of long-term fairness in the setting where the result of predicting impact the data distribution in a feedback loop. The authors extend the concept of performative prediction Perdomo et al. (2020) from purely empirical risk minimization settings to distributionally robust settings. The results show that in the long term, distributionally robust performative prediction achieves fairness even without knowing protective attributes.

**Summary Of The Review:**

Novel and sound paper with clear presentation.
Therefore, I recommend acceptance.

---

### Official Review · Reviewer_EH8v · 2022-10-29

**Confidence:** 4
**Correctness:** 3
**Technical Novelty And Significance:** 3
**Empirical Novelty And Significance:** 3
**Recommendation:** 5

**Clarity, Quality, Novelty And Reproducibility:**

The paper has a clear and smooth language. The definitions, assumptions, and backgrounds are very clear. The idea of RDRO in performative prediction is novel and interesting. What it lacks is an intuition about where the assumed setting holds and fails, which can be explained with example. Also, a clean theoretical discussion connecting why RDRO is good for the long-term fairness problem under concern is not logically evident.

**Strength And Weaknesses:**

Strength:
1. The problem of ensuring long-term fairness where the demographic information is not available, is an interesting and relevant problem.
2. The idea of doing DRO in performative prediction is an intuitive and interesting problem to study.
3. The paper presents the required definitions, assumptions, and background materials clearly and rigorously.
4. The experiments show interesting benefits of performing RDRO.

Weakness:
1. The motivation and the proposition seem disconnected to me. Though the paper begins with the problem of ensuring long-term fairness without demographic info, I do not see explicitly any proof or discussion to connect them. For example, it would be interesting to know how does it relate or contradict existing fairness metrics. Or even if we learn a \theta^t with RDRO, how much fairness violation it can induce in terms of any group fairness metric? How far or close the classifier learned with DRDO is with the fair classifiers learned using demographic information? As none of these pieces are available, it reads more like a DRO paper for performative predictions with some side-effects on fairness than a paper pointed to fairness. Please clarify if I miss any detail.
2. The definitions and assumptions are mathematically clear. But some intuition or an example to comprehend them would be very helpful.
3. The cost of RDRO wrt the performative prediction is not discussed. It would be nice to understand how hard/easy it is do RDRO than PP and what is the cost in terms of sample complexity.

**Summary Of The Paper:**

This paper studies distributional robustness in the setting of performative prediction. Specifically, it proposes to use DRO with performative prediction as a way to ensure long-term fairness where the demographic information is not available beforehand. The paper further explicates the definitions, assumptions, and constructs require to perform repeated RM in this setting. It theoretically shows that the repeated DRO in this setting should converge under given conditions. It also experimentally shows the benefits of using this approach.

**Summary Of The Review:**

I think the paper studies an interesting question and an interesting problem setup. It is cleanly written. But much remains to be discussed and shown than the present draft under considerations. It will be imperative to answer the questions in weakness to judge the value of the propositions.

---

### Decision · Program_Chairs · 2023-01-20

**Decision:**

Reject

**Justification For Why Not Higher Score:**

Proposed changes and promised experiments were not submitted for the reviewers to assess.

**Justification For Why Not Lower Score:**

N/A

**Metareview: Summary, Strengths And Weaknesses:**

Overall, the paper presents an interesting extension of performative prediction to distributionally robust settings. Such an extension is important for scenarios where the protected attributes are not known in advance. The authors present some theoretical and empirical results. All of the reviewers recognized the novelty of the introduced concepts, and found it to be interesting to the community.

Several issues have been pointed out by the reviewers. One of the key issues is an inadequate discussion of previous work and the broader literature, in particular, how the current paper places itself among this previous work. Missing work may cause scholarship issues down the road in the community. The reviewers also requested some extra experiments, which the authors said they were running and that they would appear in the updated draft but the update was never uploaded. The authors also suggested they would add some extra examples and extended discussion clarifying existing connections between RDRO and long-term fairness. In general, the reviewers were open to reconsideration of their scores if the issues had been addressed. Unfortunately, it seems like the authors promised several important changes in their rebuttals, but no new drafts of the paper had been uploaded.